# Thalamocortical synapses in the cat visual system in vivo are weak and unreliable

**Madineh Sedigh-Sarvestani\*, Larry A Palmer, Diego Contreras\***

Department of Neuroscience, School of Medicine, University of Pennsylvania, Philadelphia, United States

**Abstract** The thalamocortical synapse of the visual system has been central to our understanding of sensory computations in the cortex. Although we have a fair understanding of the functional properties of the pre and post-synaptic populations, little is known about their synaptic properties, particularly in vivo. We used simultaneous recordings in LGN and V1 in cat in vivo to characterize the dynamic properties of thalamocortical synaptic transmission in monosynaptically connected LGN-V1 neurons. We found that thalamocortical synapses in vivo are unreliable, highly variable and exhibit short-term plasticity. Using biologically constrained models, we found that variable and unreliable synapses serve to increase cortical firing by means of increasing membrane fluctuations, similar to high conductance states. Thus, synaptic variability and unreliability, rather than acting as system noise, do serve a computational function. Our characterization of LGN-V1 synaptic properties constrains existing mathematical models, and mechanistic hypotheses, of a fundamental circuit in computational neuroscience.

DOI: https://doi.org/10.7554/eLife.41925.001

## Introduction

Physiological studies of sensory thalamic nuclei and cortices have shed insight into the mechanisms of sensory encoding at successive processing stages (*Hubel and Wiesel, 1962*; *Reid and Alonso, 1995*; *Bruno and Sakmann, 2006*). Relatively less is known about how the thalamus exerts its influence on the cortex, particularly in vivo conditions. In the visual system, thalamic cells from the lateral geniculate nucleus (LGN) respond to visual input with temporally precise spikes (*Reinagel and Reid, 2000*; *Liu et al., 2001*; *Kumbhani et al., 2007*). This information would be transmitted faithfully to the primary visual cortex (V1) if thalamocortical synapses operated linearly and reliably. However, synapses that are unreliable or exhibit nonlinearities such as short-term depression, could serve to shape and limit the sensory information that reaches the cortex. Understanding dynamic properties of the thalamocortical synapse in vivo will improve our understanding of information flow from sensory thalamic nuclei to sensory cortices. However, due to the difficulty of the required experimental design, no previous studies have characterized the variability and reliability of thalamocortical synapses in vivo.

In addition, it has become clear that no more than 10% of the synapses onto thalamorecipient cortical cells in V1 are of thalamic origin (*Ahmed et al., 1998*; *Peters and Payne, 1993*; *da Costa and Martin, 2009*). However, experimental data indicate that thalamic input accounts for 1/2 to 1/3 of the activity in L4 (*Chung and Ferster, 1998*). The apparently disproportionate contribution of the thalamus to cortical response likely relies on the specific properties of the synapses involved. Although several studies in vitro (*Stratford et al., 1996*; *Cruikshank et al., 2007*; *Ohana et al., 2012*; *Kloc and Maffei, 2014*; *Cossell et al., 2015*; *Hu and Agmon, 2016*) have produced estimates of thalamocortical and intracortical synaptic strength, there is no unified conclusion on the 'robustness' of the thalamocortical connection. A seminal in vitro study (*Stratford et al., 1996*) reported powerful and invariant thalamocortical synapses in the cat, suggesting that it is the relative strength

**\*For correspondence:**
msarvestani@gmail.com (MS-S);
diegoc@upenn.edu (DC)

**Competing interests:** The authors declare that no competing interests exist.

of these synapses, compared to the more numerous intracortical synapses, that contributes to the disproportionate effect of the thalamus on V1. However, there has been no confirmation of these properties in vivo.

In this study, we used intracellular recordings in cats in vivo to characterize the dynamics of individual excitatory post-synaptic potentials (EPSPs) produced by single pre-synaptic LGN spikes driven by visual stimuli. We measured the fraction of pre-synaptic LGN spikes that produced a detectable depolarization in the cortex (reliability), the variability of the amplitude of EPSPs produced by single LGN spikes, and short-term synaptic plasticity. We then constrained a feedforward model of the thalamocortical circuit with these measured parameters and studied their effect on cortical firing and information transfer.

We found that thalamocortical synapses are highly variable and unreliable, and exhibit short-term plasticity. Our data-driven simulations showed that variable and unreliable synapses increase cortical firing and thereby boost the thalamocortical drive. This suggests that the high variability of the thalamocortical synapse, rather than being irrelevant system noise, does play a functional role in driving cortex. Our findings are consistent with recent work suggesting synaptic variability and reliability play an important role in ethologically relevant behaviors (*Evans et al., 2018*).

## Results

To study the dynamic properties of thalamocortical synapses in vivo, we recorded simultaneously from the LGN and its postsynaptic targets in L4 (n = 28) or L6 (n = 8) of V1 (*Figure 1A*). We used the same dataset as in *Sedigh-Sarvestani et al. (2017)*. All cortical cells in the database were simple cells since we found no connections with complex cells. Furthermore, we found no differences between simple cells in L4 with those in L6 and therefore the data are presented together. The goal was to record from one or more of the many pre-synaptic LGN neurons that converge onto the same target V1 neuron (*Figure 1B*). Spikes from 4 to 25 LGN neurons were recorded extracellularly with independently movable tetrodes, while the membrane potential (Vm) of V1 neurons was recorded with intracellular sharp electrodes (*Figure 1C*). Visually-aided automated clustering algorithms were used to sort the spikes into different units (*Figure 1—figure supplement 1*). This method allowed us to study the EPSPs in V1 neurons, recorded from the soma, produced by single pre-synaptic LGN spikes (*Figure 1D*, red traces). Our previous study published on this dataset (*Sedigh-Sarvestani et al., 2017*) focused on the relationship between the average connection strength and relative LGN-V1 receptive-field overlap. In this study, we focus on the dynamic properties of the LGN-V1 synapse.

### Identification of connected LGN-V1 pairs.

To identify connected LGN-V1 pairs we generated the spike triggered average (STA) of the cortical cell's Vm triggered by LGN spikes during the response to white noise stimuli (*Figure 1D*, see Materials and methods). To be considered a connection, the STA had to be above the confidence limits calculated using bootstrap techniques (Materials and methods and *Sedigh-Sarvestani et al., 2017*). Data from two experiments are shown in *Figure 1C–D*. In each case, only one of the three LGN cells was connected, as shown by the sharp, low latency, depolarization in its STA. Using this procedure, we found 36 LGN cells monosynaptically connected to 23 cortical simple cells, from a total of 296 simultaneously recorded pairs with overlapping receptive fields.

Since our previous report with this same dataset (*Sedigh-Sarvestani et al., 2017*) showed surprisingly small average EPSPs (from 0.15 to 1.2 mV, mean = 0.42 mV), here we aimed to understand the dynamics of the EPSPs triggered by single LGN spikes (single-spike EPSPs) that contribute to the average. In particular, we were interested in their amplitude variability, reliability and presence of short-term synaptic plasticity. Since true synaptic reliability, that is the variance in synaptic transmission events with release of neurotransmitter that successfully binds to receptors and causes a synaptic current flux (*Branco and Staras, 2009*), cannot be measured in vivo, here we use the term reliability simply to signify the percentage of LGN spikes that caused an observable somatic depolarization. Such somatic depolarizations are the result of the sum of true single synaptic events occurring in the dendrites of V1 cells - beyond reach to our presumably somatic intracellular recordings. The summation of such events at the soma level, which is what we measure, ultimately determines whether the Vm crosses spike threshold (*Figure 2A*).

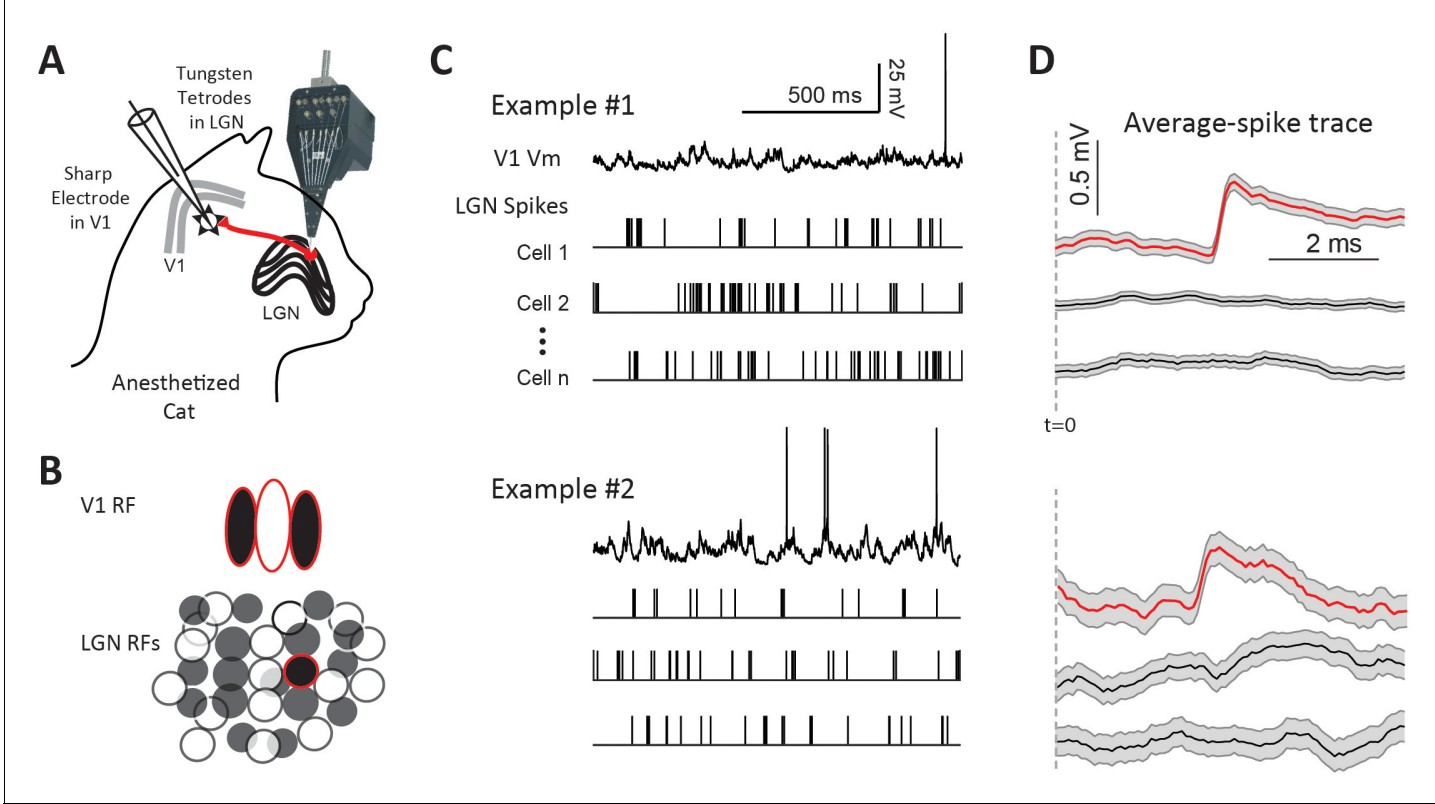

**Figure 1.** Paired LGN-V1 recordings allow quantification of average thalamocortical EPSPs in connected pairs. (**A**) Simultaneous recording of intracellular potentials with a sharp microelectrode from a single V1 neuron and extracellular potentials from many LGN neurons. (**B**) This allows us to identify one, or more, LGN cells monosynaptically connected to the cortical cell. (**C**) Sample traces, from two different example experiments, of a V1 cell together with spikes from three simultaneously recorded LGN cells, during the response to white noise. (**D**) Jitter-corrected spike-triggered average (STA) trace of V1 Vm, and SEM shown in gray, for each LGN cell. The red trace in each panel contains a sharp depolarization indicative of an EPSP from a connected LGN-V1 cell pair. The remaining LGN cells were not connected, exhibiting a flat STA. t = 0 indicates LGN spike time.

DOI: https://doi.org/10.7554/eLife.41925.002

The following figure supplement is available for figure 1:

**Figure supplement 1.** LGN cluster metrics for all connected cells used in this study.

DOI: https://doi.org/10.7554/eLife.41925.003

## Detecting Single-spike EPSPs

Examination of the Vm following each single LGN spike revealed that not all contained a sharp, short latency, depolarization indicative of an EPSP. *Figure 2B* shows all (n = 2074) single-spike traces for a sample connected pair, with the average overlaid in black. Some single-spike traces contained clear and large EPSPs (*Figure 2C* black traces), and some traces did not contain any sign of an EPSP or small depolarization (*Figure 2D*). In addition to these, it was hard to determine visually whether or not there was a small EPSP in ~5% of single-spike traces (*Figure 2C* colored traces). To remove subjective bias in categorizing spikes into those producing detectable (*Figure 2F*, top) and undetectable (*Figure 2F*, bottom) single-spike EPSPs, we developed a binary detector (Materials and methods). The lack of a detectable EPSP could be due to a variety of factors, including vesicle release failures at the thalamocortical synapse or the coincidence of transmitter release with a postsynaptic IPSP. Since, in vivo, we cannot measure those relevant transmission variables or infer the presence of a very small EPSP, we simply refer to single-spike traces without a visible EPSP as 'undetectable' single-spike EPSP. We note however that EPSPs that are so small as to be undetectable likely do not influence the post-synaptic Vm significantly, at least under our conditions, and therefore have little to no impact on the spike output of the cortical cell.

For each single-spike trace, the detector looked for the presence of a depolarization that exceeded a defined threshold within an expected 'monosynaptic window', thus imposing both an

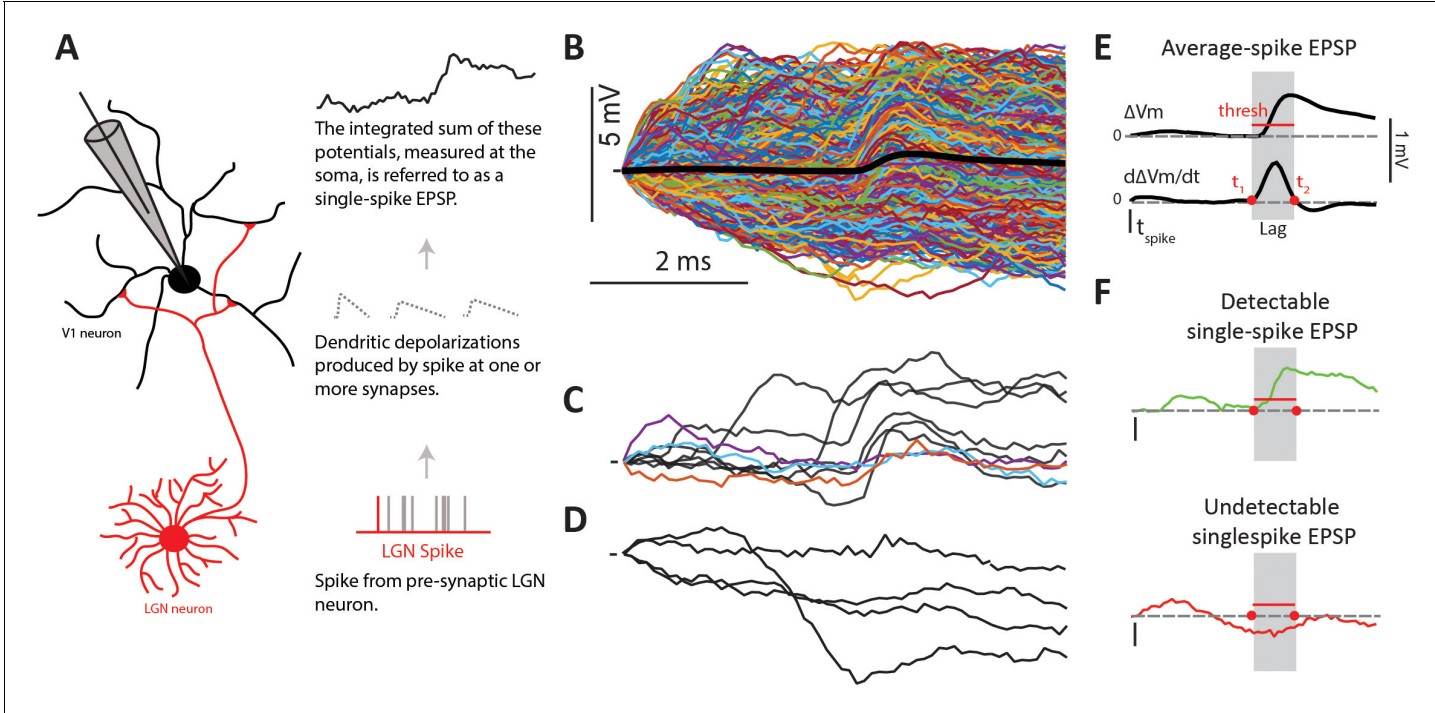

**Figure 2.** Single-spike EPSPs represent the somatic depolarization produced by a pre-synaptic LGN spike. (**A**) An LGN axon from a pre-synaptic cell makes synaptic contacts with its target L4 cell, and therefore causes dendritic depolarizations, which are integrated at the soma. We measure this integrated depolarization with a sharp intracellular electrode. (**B**) Single-spike traces (n = 2074) for a sample connected pair, overlaid with their average in black. (**C**) Subset of single-spike traces from B) exhibiting a clear EPSP (black traces), and those without a clear EPSP (colored traces). (**D**) Subset of single-spike traces from B) that lack a detectable EPSP. (**E**) The average EPSP and its first derivative are used to determine parameters for the single-spike EPSP detector (see text). The gray region indicates the 'monosynaptic window', or the expected window for a monosynaptic EPSP to occur after an LGN spike. (**F**). Example traces categorized as detectable single-spike EPSP (top) and undetectable single-spike EPSP (bottom).
DOI: https://doi.org/10.7554/eLife.41925.004

amplitude and a temporal constraint to the detection of single-spike EPSPs. Both the threshold and the monosynaptic window were defined based on the STA (*Figure 2E*), and thus unique for each connected pair. The threshold was defined as 1SD above the mean of jittered single-spike traces (*Figure 3B*, blue traces) produced by jittering the time of each spike by a random amount, 0 to 16 or 0 to 24 ms. This temporal jitter matches the duration of our stimulus and removes the effect of the pre-synaptic LGN spike, leaving only the lower frequency modulations caused by the visual stimulus. The monosynaptic window during which the single-spike EPSP had to occur was defined as the time between onset (t1) and termination (t2) of the dVm/dt of the average EPSP for that connected pair (*Figure 2E*), which corresponds with the onset and peak of the average EPSP. The monosynaptic window was imposed to reduce possible false detections of single-spike EPSPs produced by other LGN cells. Using these criteria, the detector labeled each single-spike trace as either a 'detectable single-spike EPSP' (*Figure 2F*, green) or an 'undetectable single-spike EPSP' (*Figure 2F*, red).

## Validating single-spike EPSP detector.

To validate the detector we a) analyzed the statistics of the subset of detectable and undetectable single-spike EPSPSs and b) compared the categorization of the detector to that produced by an independent linear discriminant classifier trained on a small subset of hand-labeled single-spike traces (*Figure 3—figure supplement 1*). In the example shown in *Figure 3A*, the average EPSP for the detectable group (green) has a clear sharp depolarization, whereas the average for jittered-spikes group (blue) is flat as expected. The average for the undetectable group (red) has a slight negative slope in the monosynaptic window (gray box) and no depolarization. Examination of categorized single-spike traces revealed that all traces categorized as detectable had a sharp depolarization in the monosynaptic window (*Figure 3B*, green). Traces categorized as undetectable either had

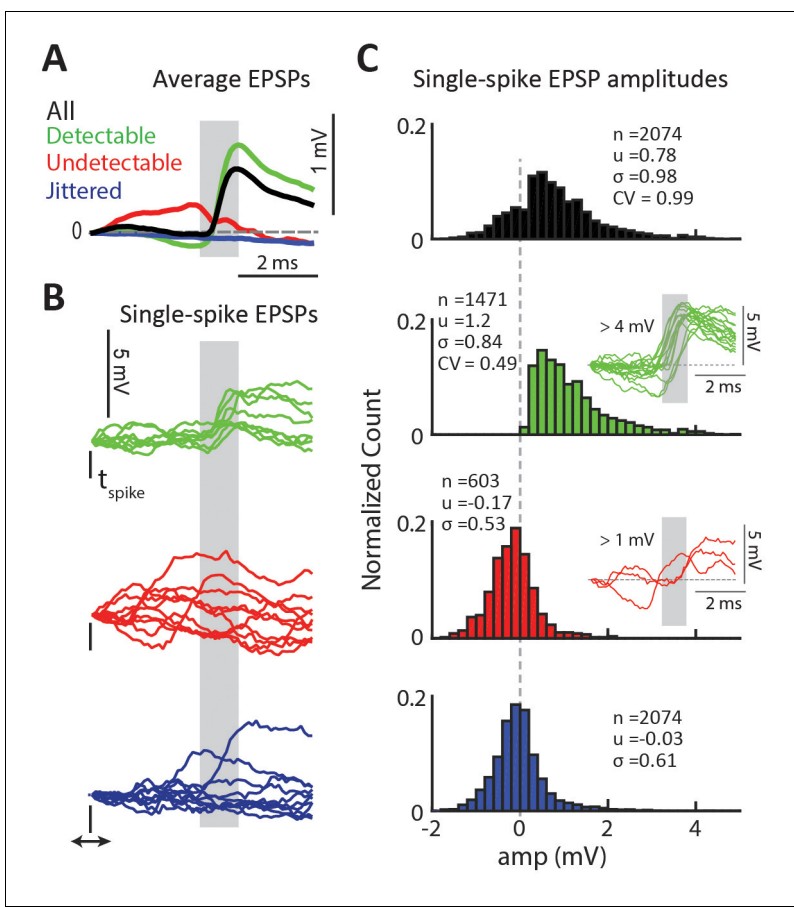

**Figure 3.** Properties of detectable and undetectable single-spike EPSPs for an example connected pair. (**A**) Average EPSP for different spike-categories within the same connected pair. Colors used throughout figure. Gray region indicates the monosynaptic window. (**B**) Example single-spike EPSPs for each category. (**C**) Distribution of single-spike EPSP amplitudes for each category. The inset in the second row shows sample detectable single-spike EPSPs with amplitudes at the high tail of the distribution,>4 mV. The inset in the third row shows sample undetectable single-spike EPSPs with amplitudes also at the high tail of the distribution,>1 mV.

DOI: https://doi.org/10.7554/eLife.41925.005

The following figure supplements are available for figure 3:

**Figure supplement 1.** Validating EPSP classification with linear discriminant analysis.

DOI: https://doi.org/10.7554/eLife.41925.006

**Figure supplement 2.** Single-spike traces for an additional sample connected LGN-V1 pair.

DOI: https://doi.org/10.7554/eLife.41925.007

**Figure supplement 3.** Less common large single-spike EPSPs contain a single sharp depolarization and have a similar shape as the more common small single-spike EPSPs.

DOI: https://doi.org/10.7554/eLife.41925.008

no depolarization or, in a smaller fraction of cases, had depolarizations that were outside of the monosynaptic window and thus were attributed to the action of other LGN inputs (*Figure 3B*, red). Consistent with the slightly negative Vm in the average of undetectable EPSPs observed for many connected pairs (*Figure 3A*, red), we found that undetectable single-spike traces often had a negative slope, which we hypothesized was due to the coincidence of an IPSP with the single-spike EPSP or to the occurrence of the spike close to the peak of an ongoing EPSP (large negative trace in *Figure 2D*). We show classified single-spike traces for a second connected pair in *Figure 3—figure supplement 2*.

## Thalamocortical synapses are unreliable and generate highly variable EPSPs.

To estimate the variability of single-spike EPSP amplitudes, we calculated the amplitude distributions for all single-spike traces (*Figure 3C*, black) as well as for the subsets categorized as detectable (green) and undetectable (red) single-spike EPSPs. As a control for the amplitude of Vm fluctuations independent of the presence of EPSPs, we also calculated the amplitude distribution for jittered-spike traces (blue). To calculate the single-spike EPSP amplitude we used the dVm/dt of the EPSP, as shown in *Figure 2E*. We refined the estimate of the onset (t1') and offset (t2') of the EPSP for each single-spike trace and defined the amplitude as ΔVm (t1':t2'). For traces without a detectable EPSP as well as for jittered traces, we defined the amplitude as the mean ΔVm across the previously defined monosynaptic window (t1:t2, based on the STA, *Figure 2E*).

These amplitude distributions suggested that single-spike EPSP amplitudes were highly variable. For the example connected pair in *Figure 3*, the distribution of all ΔVm values (n = 2074, *Figure 3C* black) was broad with a mean of 0.78 ± 0.98 mV and a range of −1.7 to 6.1 mV. The distribution of ΔVm for detectable EPSPs only (*Figure 3C*, green) had a larger mean of 1.2 ± 0.84 mV. This larger mean is due to the exclusion of traces with an undetectable EPSP, which reduce the average EPSP value. The shape of the detectable EPSP amplitude distribution had a long tail, extending from 0.18 mV to 6.1 mV. This is striking because it shows that there is a population of very large (>4 mV) EPSPs triggered by LGN input. The smooth and fast rising phase of these large EPSPs (*Figure 3C*, green traces inset), as well as similarity in shape to smaller EPSPs (*Figure 3—figure supplement 3*), suggests that they are single EPSPs and not a summation of several EPSPs. The amplitude distribution for undetectable EPSPs (*Figure 3C*, red bars) was Gaussian and centered near 0 mV, similar to the distribution for jittered traces, which represent inherent Vm fluctuations not associated with pre-synaptic spikes. Non-zero amplitudes in undetectable single-spike EPSPs were due to a positive slope in the monosynaptic window arising from an EPSP that occurred too early (from another LGN cell), or from depolarizations that did not contain a smooth slope (*Figure 3C*, red traces inset).

On average thalamocortical synapses were very unreliable. We quantified the percent of undetectable single-spike EPSPs from the total number of presynaptic LGN spikes. The mean percent of undetectable EPSPs for the population (n = 36 pairs) was 37 ± 12% [mean ± SD], ranging from 7.4% to 56% (*Figure 4A*). This large proportion of spikes that failed to produce a detectable EPSP greatly affected the amplitude and variability of the average EPSP. Across the population of connected pairs, the average EPSP amplitude was 0.49 ± 0.23 mV when including all single-spike traces (*Figure 4B*, black), and significantly higher at 0.89 ± 0.24 mV (p=3e-9, *Figure 4D*, 'A' vs. 'D') when considering only detectable EPSPs (*Figure 4B*, green). The population average amplitude for jittered spikes was close to 0 (0.01 ± 0.03) mV as expected. In contrast, the population average for undetectable EPSPs was −0.20 ± 0.09 mV, significantly different from 0 (p=5e-15 *Figure 4D*, 'U'). We speculate that this negative Vm could be due to an ongoing EPSP at the time of arrival of the LGN spike, which would lead to negative values in the calculation of its amplitude, since we zero the Vm at the time of the LGN spike. Alternatively, undetectable EPSPs could be due to the concomitant occurrence of an IPSP, which would suppress the EPSP and cause hyperpolarization. Both phenomena would lead to the slightly negative average Vm values we observe following spike failures.

We quantified the variability of the LGN-V1 synapse by calculating the noise corrected coefficient of variation (CV, Materials and methods) of single-spike trace amplitudes. In agreement with the large variability displayed by the traces in *Figure 2B*, we found large values of CV across the population, with a mean of 1.2 ± 0.47 (*Figure 4C*, black). This is nearly an order of magnitude larger than reports from in vitro studies of the cat thalamocortical pathway (*Stratford et al., 1996*) and is consistent with the high fraction of undetectable EPSPs. Indeed, the CV distribution shifted towards much lower values after removing undetectable EPSPs and its mean reduced tenfold (0.18 ± 0.13, *Figure 4C*, green), indicating that much of the variability could be explained by undetectable EPSPs.

Across the population, the percentage of undetectable EPSPs was negatively correlated with the average EPSP amplitude (r = −0.81; *Figure 4E*, left). Even though it is plausible that such negative correlation may be influenced by the amplitude threshold of the automated detector, we obtained the same relationship when using the linear classifier which has no explicit amplitude threshold. Furthermore, an inverse relationship between the rate of spike-transmission failures (analogous to our 'undetectable' EPSP rate) and average EPSP amplitude has been previously reported in a variety of

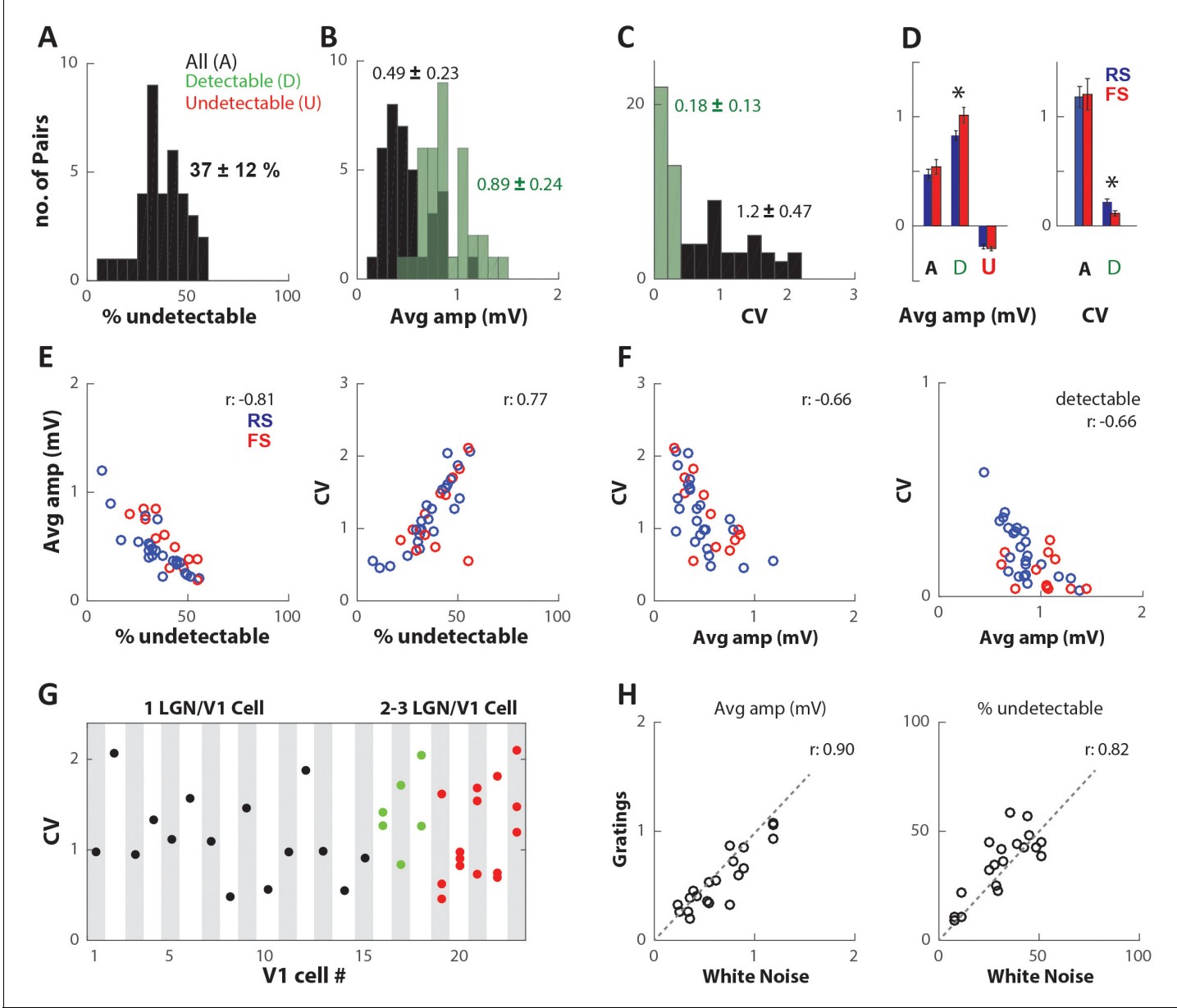

**Figure 4.** Thalamocortical synapses produce highly variable and unreliable EPSPs. (A) Distribution of percentage of undetectable single-spike EPSPs (% undetectable) for all n = 36 connected pairs. Numbers indicate mean ± SD. (B) Distribution of average trace amplitude (Avg amp), averaged over all single-spike traces (All (A), black), and only detectable single-spike EPSP traces (Detectable (D), green). (C) Distribution of coefficient of variation (CV) of single-trace amplitudes, calculated from all traces (black), and detectable EPSP traces only (green). (D) (Left) Population average (error bars indicate SEM) of Avg amp for all (A), detectable (D), and undetectable (U) single-spike EPSPs. Blue/red bars within category indicate connections onto regular spiking (RS)/fast spiking (FS) V1 cells. (Right) Population average of CV values for all spikes and detectable EPSPs. (E) (Left) Avg amp decreases with increasing %undetectable. (Right) CV increases with increasing %undetectable. Colors indicate connections onto RS or FS cells. (F) CV decreases with increasing Avg amp for all spikes (left) and for detectable only (right). (G) CV plotted for all n = 36 connected pairs, separated on the x-axis according to the target V1 cell #. Black/green/red indicates V1 cells that had 1/2/3 connected LGN cells in our dataset. (H) Avg amp and %undetectable for a subset of connected pairs during response to white noise vs. drifting gratings.

DOI: https://doi.org/10.7554/eLife.41925.009

The following figure supplement is available for figure 4:

**Figure supplement 1.** Factors that influence synaptic variability.
DOI: https://doi.org/10.7554/eLife.41925.010

circuits both in vitro (*Feldmeyer et al., 1999*, *Feldmeyer et al., 2006*) and in vivo (*Jouhanneau et al., 2015*), and is consistent with binomial models of synaptic transmission (Figure 8 in *Feldmeyer et al., 1999*). In addition, we observed a positive correlation between the percentage of undetectable EPSPs and CV (r = 0.77; *Figure 4E*, right), consistent with previous reports in vitro. The exclusion of undetectable EPSPs did not change the correlation between CV and average EPSP amplitude (*Figure 4F*, right; r = −0.66), indicating that these two values are negatively correlated independent of detectability.

To determine if synaptic reliability or variability were related with the electrophysiological class of the target V1 cell, we classified our L4 simple cells into regular-spiking (RS; n = 16/23) and fast-spiking (FS; n = 7/23) based on their intracellularly measured electrophysiological properties (Materials and methods). Of the 36 presynaptic LGN cells, 24 were connected to RS cortical cells (*Figure 4*, blue dots) and 12 were connected to FS cells (*Figure 4*, red dots). We did not find a significant difference in either average EPSP amplitude, %undetectable, or CV between RS and FS cells (*Figure 4E*). However, after excluding undetectable EPSP traces, the mean CV of RS cells (n = 24, 0.22 ± 0.13) was higher than that of FS cells (n = 12, 0.11 ± 0.08; p=0.047 KS test, p=0.018 t-test; *Figure 4D*) and the average EPSP amplitude (*Figure 4F*, x-axis) was smaller for RS (0.83 ± 0.21 mV) than for FS cells (1.0 ± 0.21 mV; p=0.0046 KStest, p=0.025 t-test; *Figure 4D*). Similarly, the average slope of single-spike EPSPs was slightly higher for FS cells (1.14 ± 0.23) compared to RS cells (0.97 ± 0.18) (p=0.023 KStest; p=0.027 t-test), only after excluding undetectable EPSPs. This suggests that undetectable single-spike EPSPSs mask differences in the size, and rate of change, of Vm fluctuations between these cortical cell types. This is consistent with previous work from our laboratory in vivo, showing different input resistance and membrane time-constant for RS and FS cells (*Contreras and Palmer, 2003*; *Cardin et al., 2007*; *Cardin et al., 2010*), and consistent with greater differences observed in vitro across these cell types (*Hull et al., 2009*; *Cruikshank et al., 2007*; *Schiff and Reyes, 2012*; *Kloc and Maffei, 2014*).

We next wanted to know whether synaptic variability was dependent on postsynaptic mechanisms. If so, LGN cells that share the same postsynaptic target should share similar CV values. Our dataset included five cortical cells with three connected LGN cells each (*Figure 4G*, red dots) and three cortical cells with two connected LGN cells each (*Figure 4G*, green dots), the remaining 15 cortical cells had a single connected LGN cell (*Figure 4G*, black dots). We plotted the CV of all connections in rank order and found that the CV of connections to the same postsynaptic V1 cell was as variable as that of connections across different cortical cells. Thus, our data suggest that the postsynaptic cell does not overtly determine the variability of LGN-V1 synapses.

To further explore postsynaptic mechanisms on EPSP variability and reliability, we estimated the correlation between instantaneous Vm (as a measure of driving force) and the single-spike EPSP amplitude for all spikes in each connected pair (*Figure 4—figure supplement 1*). As detailed in Appendix 1, we did not find a significant correlation between postsynaptic Vm and EPSP amplitude (*Figure 4—figure supplement 1F*). However, we found that spikes that occurred during hyperpolarized Vm produced EPSPs with slightly lower %undetectable rates and CVs (*Figure 4—figure supplement 1D*).

Finally, to verify that synaptic properties were not dependent on the type of visual stimulus presented, we separately calculated amplitude distributions and %undetectable for a subset (n = 19) of connected pairs during the response to oriented drifting grating stimuli (Materials and methods). Because the white noise and drifting grating stimuli were presented about ten minutes apart, this comparison also assessed the stationarity of our characterization across time. We found a high correlation between the average EPSP amplitude during the response to both white noise and gratings (*Figure 4H*, left, r = 0.90), as well as between the standard deviation of the amplitude distribution (r = 0.85, data not shown), and between %undetectable (*Figure 4H*, right, r = 0.82). We used paired t-tests to compare these response properties associated with the same synapse during presentation of the two types of stimuli. We found that the average EPSP amplitude was significantly higher during white noise stimuli (p=0.0055), but all other variables including LGN firing rate had similar distributions. The white noise stimulus has low spatiotemporal correlations relative to the oriented grating stimuli; we therefore expect that these two stimuli should induce different amounts of coordinated activity in the pre-synaptic LGN population. Thus, the larger EPSP amplitude during white noise is unlikely to be due to increased synchronized thalamic input as compared to drifting gratings. Overall, the similarity of average EPSP amplitude and %undetectable across these very different

visual stimuli suggests that synaptic strength, reliability and variability are not overtly influenced by the dynamics of the pre-synaptic LGN population as a whole and may instead reflect the properties of individual synapses.

In summary, we found that thalamocortical synapses in the cat visual system produce EPSPs with small and variable amplitudes, including a large percentage of undetectable EPSPs, thalamic spikes that either fail to produce an EPSP or produce an EPSP too small to visibly affect somatic Vm. Within our dataset, connected pairs with larger average EPSP amplitudes generally had lower %undetectable single-spike EPSPs and lower CVs. Such a relationship is consistent with expectations from binomial models of synaptic vesicle release (*del Castillo and Katz, 1954*), as well as with previous in vitro and in vivo studies (*Feldmeyer et al., 1999*; *Ohana et al., 2012*; *Jouhanneau et al., 2015*; *Pala and Petersen, 2015*). Our characterization of synaptic variability was not overtly governed by the visual stimulus or postsynaptic cell identity.

## Thalamocortical synapses exhibit short-term synaptic plasticity

Short-term synaptic plasticity (STSP) is an activity dependent change in synaptic strength that occurs on the order of tens to hundreds of milliseconds. Because STSP can directly influence the computations performed by a circuit (*Deng and Klyachko, 2011*; *Regehr, 2012*), there is some interest in understanding synaptic STSP properties in vivo. To demonstrate STSP in the thalamocortical synapse, we generated spike-triggered average EPSPs for LGN spikes preceded by interspike intervals (ISI) of different durations. The example cell of *Figure 5A* exhibited clear short- term synaptic depression (STSD) as the amplitude of EPSPs with the shortest ISIs (0–10 ms; 0.70 mV) was only 38% of the amplitude at long ISIs (>150 ms; 1.8 mV). *Figure 5B* shows another connected pair with clear STSD (0.47 mV at ISIs of 0–10 ms and 1.2 mV at ISIs of 200–500 ms, a reduction to 39% of the long ISI). Due to the limited duration of intracellular recordings, for most connected pairs we did not record enough LGN spikes to generate several ISI bins. Therefore, we grouped the ISIs in two categories, a short ISI between 0–50 ms, and a long ISI greater than 50 ms. We chose 50 ms because it was close to the median ISI for most connected pairs.

To determine if a connected pair exhibited significant STSP, we compared the cumulative distribution functions (CDFs) of single-spike trace amplitudes for short and long ISIs (*Figure 5C*, gray and blue curves). For the example connected pair illustrated in *Figure 5A*, the two distributions were significantly different (*Figure 5C*; p=1.5e-6 t-test), with long ISIs producing larger amplitudes, thus demonstrating STSD. Using this metric, we found that 20 out of our population of 36 connected pairs (55%) exhibited significant STSP.

To quantify the magnitude and sign of STSP across the population, we calculated the %change (*Figure 5F*) in average EPSP amplitude for short relative to long ISIs ((short − long)/long)x100). A negative %change indicates STSD and a positive %change indicates short-term synaptic facilitation (STSF). Of the 20 connected pairs that exhibited STSP (*Figure 5E*, both +amp bars), 17 exhibited STSD and three exhibited STSF. The %change across the 17 connected pairs exhibiting STSD was −32 ± 15%, and the %change across the three connected pairs exhibiting STSF was 24 ± 16%.

Notably, all three synapses exhibiting significant STSF were connected onto FS cortical cells (*Figure 5G*). However, three other FS cells exhibited significant STSD. This bimodality may be due to the fact that some of the cortical neurons we've classified as FS are actually SOM inhibitory neurons, which do not exhibit short-term depression (*Hu and Agmon, 2016*). RS cells only exhibited significant STSD.

The smaller single-spike EPSP amplitudes for short ISIs could be due to increased number of undetectable single-spike EPSPs at short ISIs. Thus, we compared the distribution of ISIs for detectable and undetectable single-spike EPSPs for each connected pair. For the cell shown in *Figure 5C* ISIs preceding undetectable single-spike EPSPs were shorter than those preceding detectable EPSPs (p=0.0019 t-test *Figure 5D*), consistent with a spike-history dependent decrease in the efficacy of synaptic transmission. Across the population, most (14/20) connected pairs that exhibited distinct EPSP amplitude distributions for short and long ISIs also exhibited distinct ISI distributions for detectable and undetectable single-spike EPSPs (*Figure 5E*, 'both'). Thus, we conclude that increased %undetectable at short ISIs accounts in part for the smaller observed single-spike EPSP amplitudes.

Models of synaptic transmission (*del Castillo and Katz, 1954*) predict that synapses with higher probability of release generate larger EPSPs with lower CV values and show proportionally more

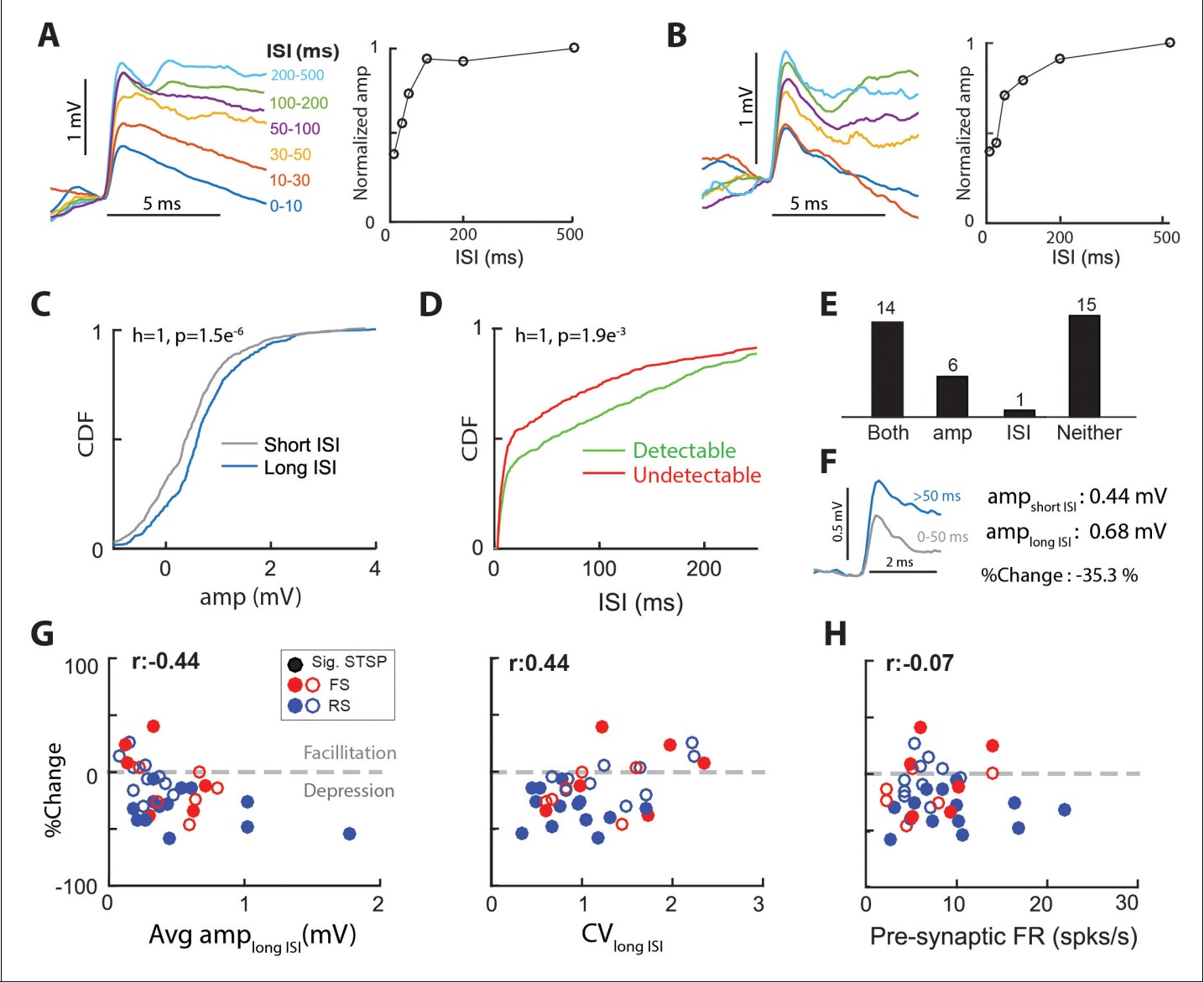

**Figure 5.** Thalamocortical synapses exhibit short-term synaptic plasticity. (**A**) (Left) Average EPSPs grouped by inter-spike-interval (ISI) for a sample connected pair. (Right) Normalized EPSP amplitude vs. ISI for the same connected pair. (**B**) Same as A, for another sample connected pair. (**C**) Cumulative distribution functions (CDF), of single-spike EPSP amplitude distribution for spikes with short (0–50 ms) ISI and those with long (>50 ms) ISI, for the connected pair in A. p values are from 2-sample KS test. (**D**) CDFs of ISI distribution for detectable and undetectable single-spike EPSPs, for the sample connected pair in A. (**E**) No. of synapses with separable distributions for both, just EPSP amp for short and long ISIs, just ISI for detectable and undetectable EPSPs, or neither. Most synapses with a separable distribution of EPSP amplitudes for short and long ISIs, also exhibited separable ISI distributions for detectable and undetectable EPSPs. (**F**) STSP was quantified as the % change in average EPSP amplitude for long-ISI EPSPs, relative to short-ISI EPSPs. (**G**) (Left) % change of average EPSP amplitude with ISI as a function of the long-ISI average EPSP amplitude for all connected pairs. Filled circles indicate significant STSP. (Right) %change plotted against CV of the amplitude distribution for long-ISI EPSPs. (**H**) %change plotted against the firing rate of pre-synaptic LGN cell.

DOI: https://doi.org/10.7554/eLife.41925.011

The following figure supplement is available for figure 5:

**Figure supplement 1.** Pre-synaptic firing rate is not correlated with synaptic variability or reliability.

DOI: https://doi.org/10.7554/eLife.41925.012

depression when activated at short intervals. Consistent with such models, we found that the connected pairs with larger average EPSP amplitudes undergo greater STSD (*Figure 5G*, left), and those with smaller average EPSPs undergo STSF (%change >0; *Figure 5G*, left). Furthermore, we also found a small positive correlation between the CV and %change (*Figure 5G*, right), with synapses with larger CVs exhibiting smaller STSD. Finally, we did not find a correlation between the amount of STSP in a connected pair and the firing rate of the pre-synaptic LGN cell within a range of 2–24 Hz (*Figure 5H*). This suggests that, at least for the range measured here, STSD should not dissipate (*Boudreau and Ferster, 2005*).

All together, we conclude that a) thalamocortical synapses exhibit STSP, with FS cells exhibiting STSD and STSF equally and RS cells exhibiting only STSD; b) at the single synapse level, spiking history influences the detectability of the EPSP and thus the influence on postsynaptic spike output; c) STSD in vivo is independent of pre-synaptic firing rate, and is associated with EPSP amplitude reductions of ~30%.

## Variable and unreliable synapses boost the thalamocortical drive in a model V1 cell.

To understand the functional implications of synaptic variability and reliability that we measured in vivo, we implemented a biologically constrained leaky integrate-and-fire (LIF) model of a V1 L4 simple cell driven by the prerecorded spiking activity of up to 50 LGN cells responding to white noise visual stimulus (*Figure 6A*, Materials and methods). The prerecorded LGN cells were chosen so that the spatial alignment of their RFs generated an elongated simple-like RF in the V1 model cell (*Figure 6B*). LGN spikes triggered transient excitatory conductances in V1 with kinetic and synaptic parameters matching our measurements in vivo.

We created two models to explore the effect of synaptic variability and reliability on V1: one model was built with reliable synapses that convert every pre-synaptic spike into an EPSP (0% undetectable) of constant amplitude ('reliable', *Figure 6C*). Another model was built with unreliable synapses with varying percentages of synaptic failures, which lead to undetectable EPSPs, ('unreliable', *Figure 6C*) and EPSPs with variable amplitude. We calibrated these two models so that they generated the same average excitatory conductance amplitude (*Figure 6C*, bottom), allowing for a controlled comparison of the effect of synaptic reliability and variability. For the unreliable model, synaptic failures were randomly selected from each LGN cell's spike-train (*Figure 6D*). All synaptic properties, the average and variance of EPSP amplitudes and failure rate (which corresponds to % undetected), were chosen from our measured dataset. Corresponding synaptic properties were kept consistent, so that synapses with larger EPSP amplitudes exhibited lower failure rate, similar to the lower % undetectable in vivo. We assigned synaptic properties randomly to each pre-synaptic LGN cell and ran 10 trials with a different randomized assignment in each trial. Adding STSD as measured in vivo did not change our results (*Figure 6—figure supplement 1*), so for simplicity STSD is excluded as a variable.

*Figure 7A* shows samples of synaptic inputs and cortical outputs for both models, when driven by 50 LGN neurons. In the reliable model each LGN spike (black dots) generated an excitatory postsynaptic conductance (Gin) of equal amplitude for each LGN cell. In the unreliable model there were random synaptic failures (*Figure 7A*, red circles) and each Gin varied around the mean value for each LGN cell. The total synaptic conductance was converted into an input current (Iin) which produced Vm depolarizations, some of which reached spike threshold (red dotted line). In this example, only the V1 cell with unreliable synapses fired action potentials, due to the presence of larger fluctuations in its Iin.

The cortical response produced by the reliable and unreliable synapses was qualitatively similar, as shown by the pattern of V1 spiking (*Figure 7B*). In addition, reverse correlation of the model Vm with the white noise stimulus showed similar simple cell-like RF subregions for both models (*Figure 7C*). However, the clear difference between the two models was the increased cortical firing rate produced by unreliable synapses (*Figure 7B*). Both model cells exhibited depolarization as a linear function of the number of LGN synapses as expected (*Figure 7E*). Since the total amount of injected conductance was kept the same between models, the average Vm depolarized equally for both. In addition, the average Vm remained below the spike threshold of −55 mV even when driven by the maximum number of LGN inputs. Thus, the spike output was entirely driven by fluctuations of the Vm, which was quantified by the standard deviation (Vm SD; *Figure 7E*). The variable EPSP

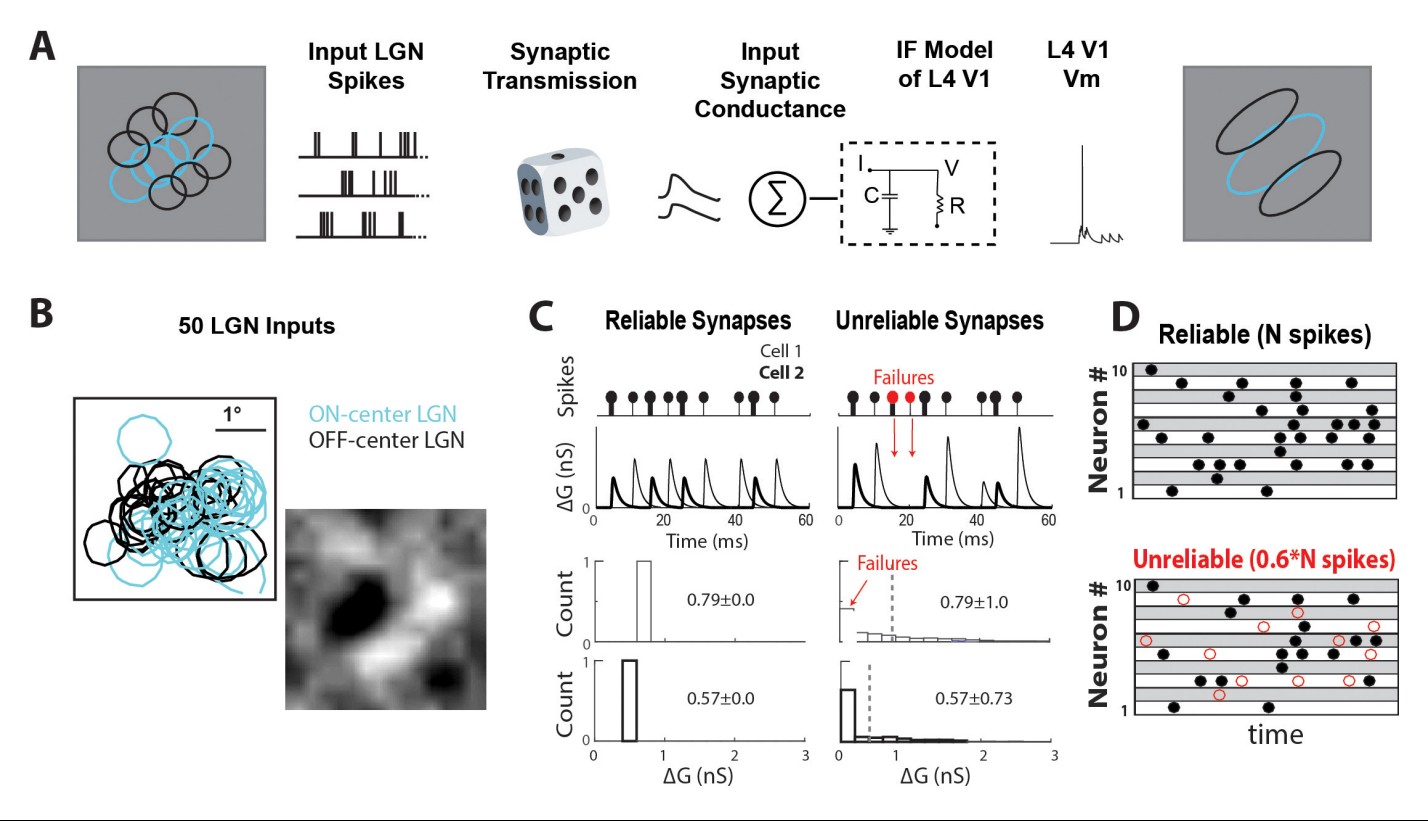

**Figure 6.** Data-driven feedforward model used to explore effect of thalamocortical synaptic variability and reliability in V1. (**A**) Model input:output schematic. Spiking responses, during white noise visual stimulus, of a group of prerecorded LGN cells selected to form a V1-like RF are used as pre-synaptic input for the model. These pre-synaptic inputs undergo either reliable or unreliable synaptic transmission, and are summed to produce a total synaptic conductance to the V1 cell. (**B**) RF-center contours of the 50 ON and OFF-center LGN cells used as inputs to the model, along with the population RF of all spikes. (**C**) (Top) Conversion of pre-synaptic spikes into excitatory post-synaptic conductance (EPSG) for model cell with two example reliable (left) and unreliable (right) synapses. (Bottom) Distribution of EPSG amplitudes for a sample synapse under reliable and unreliable model. The distributions have the same mean, but different variance. (**D**) Schematic showing random selection of ~40% of pre-synaptic spikes designated as undetectable EPSPS (failures, red circles) for each synapse in unreliable model.

DOI: https://doi.org/10.7554/eLife.41925.013

The following figure supplement is available for figure 6:

**Figure supplement 1.** Feedforward model with depressing synapses produces similar results as model without depressing synapses.

DOI: https://doi.org/10.7554/eLife.41925.014

amplitude, built-in the unreliable model, produced proportionally larger Vm SD then the reliable model as the number of LGN inputs increased (*Figure 7E*, middle). The larger VmSD in turn led to increased threshold-crossings and increased firing-rate (*Figure 7E*, right). Thus, Vm fluctuations (unreliable model) more efficiently drove cortical output than the temporal summation of constant amplitude EPSPs (reliable model). For a V1 cell driven by 50 LGN neurons with unreliable synapses, the average V1 firing rate was 7.8 ± 1.4 spikes/sec whereas for reliable synapses it was 1.8 ± 0.84 spikes/sec.

It should be noted that the difference in Vm SD produced by reliable and unreliable synapses is rather small (*Figure 7E*, middle panel), on the order of 1 mV. However, due to the exponential relationship between Vm SD and firing rate (*Figure 7D*, left), small increases in Vm SD can lead to large increases in spike output. In addition, this relationship demonstrates that Vm SD, and not mean Vm, determines cortical firing rate for both reliable and unreliable synapses (*Figure 7D*, right).

To confirm our observations, we ran several control simulations. First, to determine whether failures or EPSP variability have a greater effect on increased cortical firing rate in the unreliable model, we produced a model with unreliable synapses that had the same %failure but single-spike EPSPs with constant amplitude (*Figure 7—figure supplement 1*). We found that the effect of EPSP

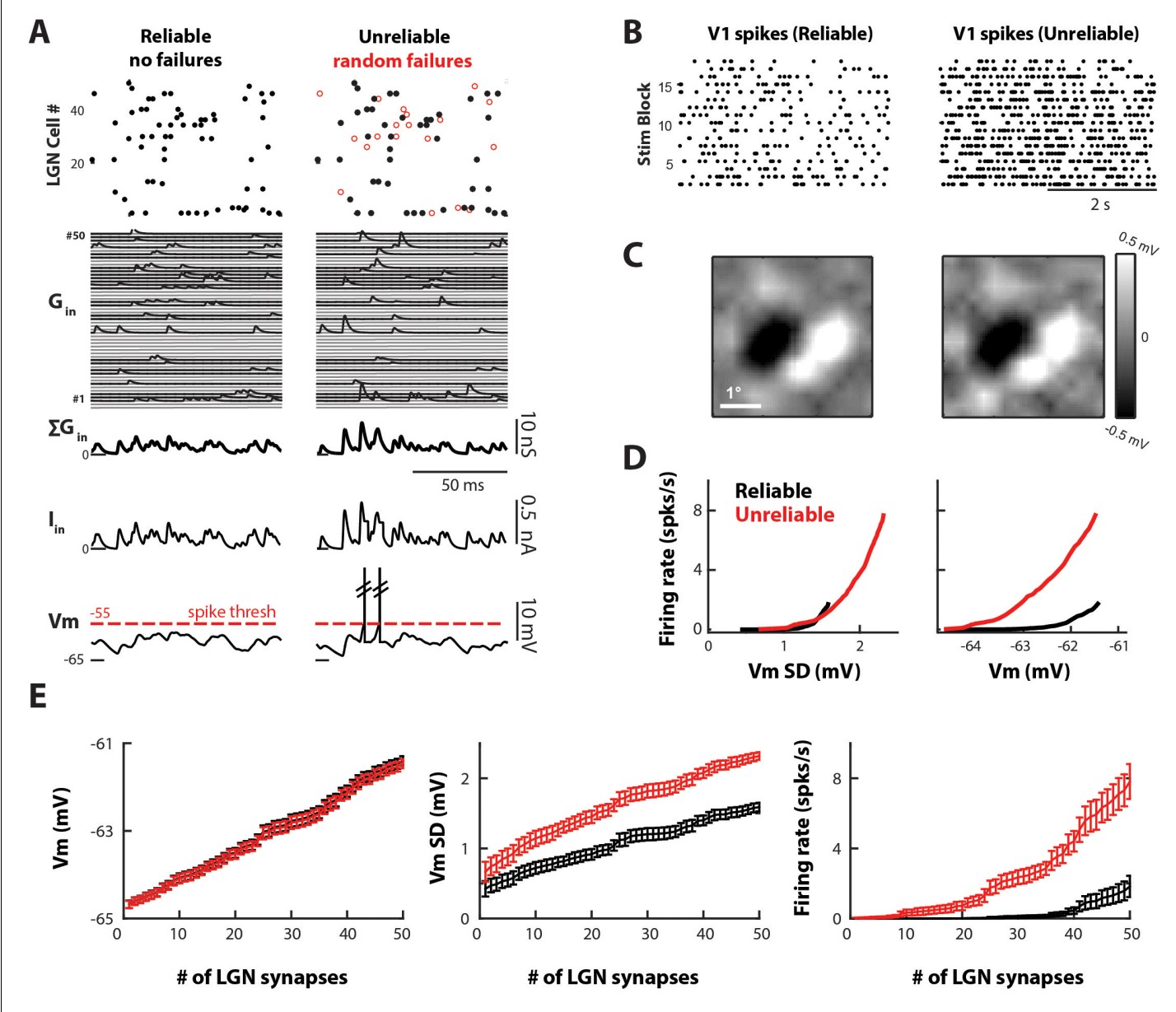

**Figure 7.** Variable and unreliable synapses boost the thalamocortical drive in a model V1 cell. (**A**) Sample 100 ms simulation for the reliable (left) and unreliable (right) model. Undetectable EPSPs are called failures. (Top) LGN Spike inputs for each of 50 synapses, with spike failures shown as red dots. (Middle) Excitatory conductance trace for each synapse, total synaptic conductance, and total current input. (Bottom) Corresponding V1 Vm, which is below spike-threshold for both models. (**B**) V1 spike rasters for 20 4 s blocks of the white noise stimulus for reliable and unreliable model. (**C**) Model V1 RFs, at time of pixel with strongest response (75 ms). (**D**) V1 firing rate as a function of V1 Vm standard deviation (Vm SD, left) and mean (right). Each point represents the average value across n = 10 simulations. (**E**) Average Vm, Vm SD, and firing rate of V1 cell as a function of the number of synaptic inputs, for n = 10 simulations of the reliable and unreliable model. Error bars indicate 95% confidence intervals.

DOI: https://doi.org/10.7554/eLife.41925.015

The following figure supplements are available for figure 7:

**Figure supplement 1.** Synaptic variability and unreliability both contribute to increased thalamocortical drive.

DOI: https://doi.org/10.7554/eLife.41925.016

**Figure supplement 2.** Unreliable synapses with non-random correlated failures also drive cortex better than reliable synapses.

DOI: https://doi.org/10.7554/eLife.41925.017

**Figure supplement 3.** Feedforward model with high conductance state created by adding balanced excitatory and inhibitory background conductance.

DOI: https://doi.org/10.7554/eLife.41925.018

variability is larger than the effect of failure rate, although we recognize that EPSP variability and reliability are not independent in vivo. Second, we changed the distribution of failed LGN spikes in the unreliable model, from random assignment as in *Figure 7* to correlated failures (*Figure 7—figure supplement 2A*), where a pre-synaptic LGN spike is more likely to fail if it occurs with spikes from other pre-synaptic LGN spikes targeting the same V1 cell. We found that the unreliable model with correlated failures (*Figure 7—figure supplement 2B*, blue traces) still produced higher cortical VmSD and firing. In summary, our simulation results show that the variability and reliability of thalamocortical synapses lead to a more efficient driving of their postsynaptic cortical cells compared to reliable synapses with constant amplitude input.

## Discussion

We used paired in vivo recordings of LGN and L4/L6 of V1 to investigate dynamics of synaptic transmission in the thalamocortical pathway of the cat visual system in vivo. In contrast to previous findings in acute slices of visual cortex, our data show that thalamocortical synapses are weak and unreliable, and exhibit synaptic depression. Furthermore, our data show that the mean amplitude, the variability and reliability of the mean EPSP triggered by single geniculocortical neurons in vivo strongly depends on the % detectability at the postsynaptic cortical cell soma. Functionally, our data-driven simulations show that this variability and reliability actually enhances cortical firing via an increase in cortical membrane potential fluctuations., that is cortical firing rate is driven by fluctuations of Vm rather than mean Vm, similar to high conductance states (*Destexhe et al., 2003*) as observed in primary visual cortex of non-anesthetized primates (*Tan et al., 2014*).This suggests that thalamocortical synaptic variability has a downstream cortical effect and serves a computational purpose. In providing key parameters of the thalamocortical synapse, our work allows for improved understanding of early sensory encoding. Furthermore, in agreement with previous work in the rodent thalamocortical pathway in the somatosensory system, our work challenges the notion of a strong and reliable thalamocortical synapse in the visual system of a highly visual animal.

### Thalamocortical strength, reliability, and variability in vivo

To our knowledge, this is the first in vivo report of thalamocortical synaptic variability and reliability. The average EPSP amplitude, the fraction of undetectable single-spike EPSPs, and the coefficient of variation that we measured together indicate weak and unreliable thalamocortical transmission. This differs drastically from findings in slices of the cat thalamocortical pathway (*Stratford et al., 1996*; *Bannister et al., 2002*), which indicated that thalamocortical connections had large amplitudes (~2 mV) with little variability (CV ~0.1). The differences in our in vivo results may be due to differences in external calcium concentrations (*Borst, 2010*), or from differences in shunting conductances (*Urban-Ciecko et al., 2015*). However, the EPSP amplitudes that we measured (~0.9 mV, mean including only detectable) are comparable to in vivo measurements of thalamocortical EPSP amplitude in the rodent L4 barrel cortex calculated by either STA from simultaneously recorded VB spikes (*Bruno and Sakmann, 2006*); 0.9 mV) or from minimal electrical stimulation in the white matter (*Schoonover et al., 2014*); 1.0 mV).

The smaller EPSP amplitudes recorded from barrel cortex in vivo already suggested that synapses in vivo would have correspondingly higher variability and failure rates, although these hypotheses were not confirmed. Furthermore, it remained unknown whether the geniculocortical pathway of a highly visual animal would also exhibit weak and unreliable transmission. Our simultaneous recordings of pre and post-synaptic cell pairs during the response to visual stimulus allowed us to characterize synaptic variability and reliability using visually driven responses, without the need for electrical stimulation that may drive multiple cells or drive single cells in an unnatural regime. We were particularly surprised at the high fraction of spikes that produced undetectable single-spike EPSPs, which was much higher than previously reported values in the slice (*Stratford et al., 1996*). There are no published reports of in vivo thalamocortical synaptic failure rates for comparison, but our average %undetectable rate of ~40% is double the value reported from in vivo single-cell stimulation studies in mouse L2/3 excitatory intracortical connections (*Jouhanneau et al., 2015*). As shown by our modeling simulations, such high synaptic unreliability influences subthreshold response and information flow across the thalamocortical network, and must therefore be taken into account in computational models.

Experimental constraints did not allow us to characterize intracortical synapses. So one limitation of this study is the lack of side-by-side comparison of thalamo- and cortico-cortical synaptic strength, as conducted in a tour de force study in the rat barrel cortex (*Schoonover et al., 2014*). However, we note that our conclusion of weak thalamocortical EPSPs is consistent with anatomical studies in the cat thalamocortical pathway that failed to find structural differences suggestive of relatively stronger thalamocortical, compared to corticocortical, synapses (*da Costa and Martin, 2011*). We also cannot make a side-by-side comparison of thalamo and cortico-cortical synaptic reliability or variability. Additional studies in highly visual animals are needed to directly compare thalamocortical and intracortical synaptic properties.

Another limitation of this study is that our measurements did not allow us to identify true vesicle transmission failures. We therefore defined reliability as the % of detectable single-spike EPSPs, that is whether or not the pre-synaptic spike produced a detectable sharp and low-latency depolarization in the post-synaptic Vm recorded from the soma. In studies of synaptic transmission, reliability is defined by the variance in synaptic transmission due to the probability of release of neurotransmitter, which successfully binds to receptors and causes a synaptic current flux and a postsynaptic potential. Such measurements are not currently possible in vivo. Furthermore, postsynaptic factors, such as shunts caused by IPSPs, may also cause variability in EPSP amplitude as well as the failures to record it in the soma. Despite these important caveats, given the most common location of the axon initial segment in the soma, it is ultimately the effect of a synaptic input on somatic Vm that influences the postsynaptic spike output. Several trends in our results suggest that the spikes classified as producing an undetectable single-spike EPSP did fail to significantly impact the post-synaptic cell and represent a valid estimate of underlying unobservable synaptic phenomena. First, the shape and mean value of the amplitude distribution for undetectable single-spike EPSP traces (*Figure 3C*, red) is similar to that for jittered traces (*Figure 3C*, blue), indicating that undetected EPSPs do not represent significant deviation from inherent membrane fluctuations. Second, synapses with larger average EPSPs have lower % undetectable EPSPs (*Figure 4E*), and this is not a byproduct of our classifier as the amplitude threshold for detectable EPSPs is independent of the STA amplitude, but is consistent with expectations from binomial models of synaptic vesicle release (*del Castillo and Katz, 1954*). Third, in some connected pairs, the % of undetectable EPSPs was associated with shorter average ISIs than detectable EPSPs (*Figure 5D,E*), consistent with depressing synapses. Nonetheless, further studies using subcellular manipulation techniques are necessary to understand the underlying sources of synaptic detectability in the thalamocortical synapse.

## Short-term synaptic plasticity in vivo

We found that slightly more than half of our thalamocortical synapses exhibit short-term synaptic depression (STSD) with a reduction of the average EPSP amplitude by ~30% across a ~ 100 ms timescale. This is consistent with previous reports of synaptic depression in the thalamocortical pathway of the anesthetized cat carried out by *Boudreau and Ferster (2005)*. They found depression rates of 20% for electrical stimulation of the LGN at 20 Hz, and of 40% for the highest stimulation rate of 100 Hz. Our depression metric is the amplitude ratio between LGN spike intervals longer and shorter than 50 ms, which lumps together their three stimulation frequencies and it so happens that our 30% ratio is right in the middle of their reported range (20% to 40%). That study suggested that in the awake animal, STSD would be undetectable due to higher spontaneous firing rates which would render the synapses constantly depressed. However, we find the degree of STSD to be independent of pre-synaptic firing rate within our range of 2 to 24 Hz (*Figure 5H*). Thus, our results suggest that, at least for firing rates within our range, STSD should be observed in the waking state. We also note that none of the synaptic parameters that we measured exhibits a relationship with pre-synaptic firing rate (*Figure 5H*, *Figure 5—figure supplement 1*), suggesting that higher thalamic firing rates would not overtly impact our results on synaptic variability or plasticity. However, we cannot be sure that the various changes in neuromodulatory concentration that come with the awake state will not impact synaptic dynamics. Further studies that characterize the same circuit across anesthetized and awake conditions (*Durand et al., 2016*) are needed.

Finally, one important caveat of this study is the use of anesthesia. Anesthesia is essential in order to achieve mechanical stability for high quality intracellular recordings, as well as to paralyze the eyes with neuromuscular blockers in order to obtain high resolution RF maps of LGN and V1 cells. We used barbiturate anesthesia for some of the experiments and propofol for others and because

the results were not different, we combined the data. Both barbiturates and propofol act by potentiating inhibitory GABAA responses (*Nicoll et al., 1975*; *Veintemilla et al., 1992*). Many of the classical vision studies in cats and primates were done under barbiturate anesthesia and neuromuscular blockade (*Kuffler, 1952*; *Hubel and Wiesel, 1962*; *Hubel and Wiesel, 1968*; *Enroth-Cugell and Robson, 1966*). The majority of such findings have been subsequently validated in non-anesthetized animals (*Wurtz, 1969*; *Read and Cumming, 2003*; *Duffy and Hubel, 2007*). Much less is known about the action of propofol in visual responses and thalamocortical synaptic transmission. However, the spatiotemporal properties of LGN and V1 cells reported here, were not different between anesthetics. Therefore, we argue that our results are not overtly altered by the use of anesthetics.

### Effect of thalamocortical variability on L4

Our simulations show that incorporating in vivo-like thalamocortical synaptic dynamic properties leads to increased cortical firing in L4, due to the increased fluctuations (VmSD) produced by synapses with variable and unreliable transmission. Both the reliable and unreliable models were simulated in a fluctuation driven or high-conductance state (*Destexhe et al., 2001*), a state observed in many cortical neurons wherein firing mainly occurs due to fluctuations in the Vm, and not mean Vm, which on average sits below spiking threshold (*Figure 7A*). In such a state, any increase in Vm fluctuations (VmSD) will lead to supralinear increases in firing rate (*Figure 7D*). We simulated the fluctuation-driven regime either with a low membrane time-constant (*Figure 7*) or by explicitly adding balanced excitatory and inhibitory background conductance (*Figure 7—figure supplement 3*, Appendix 1), and both simulations produced similar results. Our data show that synaptic variability effectively enhances the thalamocortical drive in large part because L4 neurons operate in the high-conductance state. Our results are consistent with a large number of previous studies exploring the effect of various forms of synaptic noise on gain modulation (*Ho and Destexhe, 2000*; *Chance et al., 2002*; *Cardin et al., 2008*), and specifically point to single-spike synaptic variability as a potential mechanism for gain control in the thalamocortical pathway. Further studies that specifically manipulate synaptic variability are needed to test such hypotheses.

### Conclusion

Our study contributes to the understanding of the disproportionate effect of thalamic synapses on V1 neurons by carefully characterizing the properties of thalamocortical synapses. Our results dispel the notion of a strong thalamocortical transmission in the visual system of higher mammals, but suggest that the properties we report actually enhance the thalamic drive. The next critical step in this line of inquiry, which is feasible given new genetic technologies, is to test the hypothesis that thalamic population synchrony amplifies the thalamocortical drive.

## Materials and methods

Experiments were conducted according to guidelines of the NIH and with the approval of the Institutional Animal Care and Use Committee of the University of Pennsylvania. This study uses the same dataset first reported in *Sedigh-Sarvestani et al. (2017)*, which details the surgical and experimental procedures summarized here.

### Surgery and anesthesia

Adult male cats (2.5–3.5 kg) were anesthetized with an intraperitoneal injection of Nembutal (25 mg/kg), and paralyzed with gallamine triethiodide (Flaxedil) and artificially ventilated (end tidal $CO_2$ held at 4.0%). Anesthesia was maintained by continuous infusion of sodium thiopental (3–10 mg/kg/h) for some experiments (n = 40), and propofol (3–4 mg/kg/hr) for others (n = 4). A desynchronized pattern of EEG without predominant slow oscillations, but with occasional spindles (*Steriade et al., 1993*), was maintained throughout the experiment. Atropine solution and phenylephrine HCl were administered to the corneas, to dilate the pupils and retract the nictitating membrane, respectively. Spectacle lenses were chosen by the tapetal reflection technique to optimize the focus of stimuli on the retina, and the screen coordinates of the area centrali from both eyes were recorded.

## Paired recordings

Two craniotomies and durotomies were made in the left hemisphere, at Horsley–Clarke coordinates [A6, L8] for the dorsal LGN (lamina A), and [P4, L2] for V1. The stability of the recordings was improved by bilateral pneumothorax, draining the cisterna magna, suspending the hips, and filling the cranial defect with 3.5% agar. Glass pipettes (tip resistance between 50–90 MΩ), filled with 3M potassium acetate, were used for intracellular recordings in V1. L4 simple cells were consistently found between 500 and 950 um, as measured from the micropositioner. Recordings were digitized at 33 KHz (Neuralynx) and downsampled to 10 KHz. All cortical cells used in this study had a stable Vm below −60 mV, without holding current, had overshooting action potentials, and were observed for at least 10 min. Extracellular recordings in the dorsal LGN were made with an array of tungsten in glass tetrodes (Thomas Recording GmbH), usually between 11,000–13,000 um.

To increase the likelihood of finding monosynaptically connected pairs, the V1 and LGN recording electrodes were positioned in corresponding retinotopic locations. Typically, the LGN RFs were mapped first to obtain an estimate of their projections in V1 using published anatomical maps. All of the tetrodes were moved together if the LGN RFs were estimated to map more than 5° from the right area centralis.

## Spike clustering

Spike waveforms from each tetrode in the LGN were clustered into individual units (Sedigh-Sarvestani et al., *Figure 1D*), online and offline as necessary, using a mixture of algorithmic and manual sorting (SpikeSort3D, Neuralynx). All clusters with spikes in the 0–1 ms bin (the estimated refractory period) of the interspike interval (ISI) histogram were strictly rejected. Cluster metrics are shown in *Figure 1—figure supplement 1*. The RF of all LGN clusters included in this dataset resembled the expected center-surround structure of either ON or OFF sign.

## Visual stimulation

Stimuli were presented on an Image Systems model M09 LV monochrome monitor operating at 125 frames per second, a spatial resolution of 1024 × 786 pixels, and a mean luminance of 47 cd/m2. Unless otherwise specified, thalamocortical responses were characterized during the presentation of white noise stimulus. The noise was 16 × 16 pixels in space, covering 3–5 square degrees in space, with frame durations of 8–16 ms. Each RF pixel could take one of four values from white to black. Grating stimuli consisted of drifting bars with sinusoidal variation in contrast, with temporal frequency of 2–3 Hz and spatial frequency optimized for the V1 cell.

## Electrophysiological classification of V1 cells

We classified the 23 connected V1 cells in our dataset into RS (n = 16) and FS (n = 7) types, based on their intracellularly measured electrophysiological properties (*Contreras and Palmer, 2003*). FS cells responded to current injection with non-accommodating trains of high-frequency firing of up to 300 Hz (Figure 5A in *Sedigh-Sarvestani et al., 2017*), and had smaller spike widths (<0.8 ms, measured at the base) with pronounced after-hyperpolarizations (AHP; Figure 5B in *Sedigh-Sarvestani et al., 2017*). In contrast, RS cells showed lower frequency firing, accommodating trains in response to current injection, longer duration action potentials (>0.8 ms), and absence of a clear AHP.

## Spike-triggered average (STA) to reveal connected pairs

We identified connected pairs based on the STA of the cortical Vm. Details and data on identifying connected pairs in our dataset, as well as the bootstrap procedure to test for the significance of monosynaptic EPSPs, are given in *Sedigh-Sarvestani et al. (2017)*. Briefly, STAs were calculated, for all simultaneously recorded LGN-V1 cell pairs, by averaging ±10 ms windows of cortical Vm centered on each LGN spike during responses to white noise visual stimulation (*Figure 1C–D*). To correct for common stimulus modulation independent of a monosynaptic connection, a temporal jitter was applied to LGN spike times and the STA was recalculated. The amount of jitter was sampled from a uniform distribution matching the frame rate of the stimulus (0–16 or 0–24 ms). This time scale preserves the slower modulation of the Vm caused by the visual stimulus but eliminates the faster monosynaptic EPSP. By subtracting this jittered-STA from the raw STA we obtain the jitter-corrected STA. Jitter-corrected STAs showing an EPSP in the monosynaptic window between 1.0 to 4.5 ms

(*Reid and Alonso, 1995*) and a 10–90% rise time of less than 2 ms (*Tanaka, 1983*; *Bruno and Sakmann, 2006*) were selected for significance analysis. To reach significance and be included in the database of connected pairs, the jittered-corrected STA must have more than 10 consecutive points above the 95% confidence limits calculated using bootstrap methods (*Sedigh-Sarvestani et al., 2017*). In connected pairs, the jitter-corrected STA reveals a sharp depolarization characteristic of a monosynaptic EPSP (*Figure 1D*).

## Categorizing single-spike EPSPs with automated detector

Visual inspection of the Vm traces following each LGN spike (single-spike EPSP) revealed that not all contain a detectable sharp, low latency, depolarization indicative of an EPSP. We developed a binary detector (*Figure 2E–F*) to distinguish between detectable EPSPs, which exhibit a clear depolarization (*Figure 2F*, green), and undetectable, which do not (*Figure 2F*, red). The detector is applied to single-spike traces from 0 (time of the LGN spike) to 6 ms and involves a stepwise process: (1) We calculate the dVm/dt of the jitter-corrected STA smoothed by a three point average (see above; *Figure 2E*). All dVm/dt calculations are differences between neighbor samples. (2) We search for a positive peak using the findpeak algorithm in Matlab. This produces an unequivocal single peak in every LGN-V1 synaptic connection. (3) Starting from the peak we march backwards to detect the EPSP onset (valley before peak, t1) and forward to detect the time of termination (valley after peak, t2). We define the time between t1 and t2 as the 'expected monosynaptic window' for that LGN-V1 pair. This expected monosynaptic window is imposed to reduce contamination of EPSPs by other pre-synaptic LGN cells, based on the assumption that the jitter in the latency of single-spike EPSPs produced by the same LGN cells is smaller than the difference in the average latency between two different LGN cells. For example, note that some single-spike Vm traces containing sharp depolarizations are categorized as undetectable because the depolarization occurs outside of the monosynaptic window (*Figure 3B*, red traces). (4) Before applying the detector to single-spike traces, we subtract the Vm value at the time of the LGN spike (t = 0 ms), so that all single-spike Vm traces start at 0 mV. (5) We calculate the dVm/dt for each single-spike Vm trace and (6) search for a peak within the expected monosynaptic window using the same strategy as described above for the STA. If no peak is found the trace is considered an undetectable EPSP. (7) If a peak is found we recalculate t1 (onset) and t2 (end) for the dVm/dt of the single-spike Vm trace and obtain t1' and t2'. (8) We measure the amplitude of the candidate EPSP as the Vm at t2' minus the Vm at t1'. (9) Finally, to be classified as a detectable single-spike EPSP, the calculated amplitude has to cross a threshold (*Figure 2E*) of 1 SD above the mean of the jittered spike-triggered Vm (*Figure 3B*, blue). For the example cell shown in *Figure 2E* (red line), the threshold was 0.15 mV. The average noise threshold across the population of connected pairs was 0.22 ± 0.05 mV. It is important to note that this threshold varied somewhat for each connected pair. We chose the variable threshold because the sizes of the average EPSPs varies across the 36 connections making it difficult to set a fixed threshold that makes sense for all cell pairs. However, setting a fixed threshold across all cell-pairs did not impact our results or conclusions. We visually verified the performance of the binary classifier for a subset of single-spike traces in each connected pair and generally found <5% misclassifications. Nearly all misclassifications were false-negatives, undetectable single-spike traces that were falsely classified as detectable single-spike EPSPs due to the shape of the single-spike trace which produced a positive amplitude value over the monosynaptic window despite the lack of a sharp depolarization (*Figure 3C*, inset with red traces). These misclassifications were corrected and included in the analysis. Furthermore, the similarity in the distribution for undetectable EPSPs and jittered spikes is evidence that the classifier correctly identified undetectable single-spike EPSPs (*Figure 3C*).

## Calculating coefficient of variation (CV) for each connected pair

For each connected pair, we calculated the noise-corrected CV based on the distribution of single-spike trace amplitudes (including undetectable EPSPs), corrected by the jittered single-spike trace amplitude distribution (*Feldmeyer et al., 1999*):

$$CV = \frac{\sqrt{\left|\sigma_{all}^2 - \sigma_{noise}^2\right|}}{\mu_{all}} \qquad (1)$$

Where, $\mu_{all}$ and $\sigma_{all}^2$ are the average and variance of the single-spike trace amplitude distribution

for all spikes (*Figure 3C*, black), and $\sigma_{noise}^2$ is the variance of the jittered-spike distribution (*Figure 3C*, blue) representing Vm fluctuations not related to the presynaptic LGN cell of interest.

## Generating a leaky integrate-and-fire (LIF) model of a V1 neuron

We used a LIF model of a simple cell in L4 V1, driven by the spikes of 1–50 LGN neurons responding to a white noise visual stimulus (*Figure 6A*). The white noise stimulus is the same used to map receptive fields and to find connected pairs. It consists of 16 × 16 pixels in space, covering 4° x 4°, with frame duration of 16 ms. Each pixel is one of four contrast values from black to white. Frames are organized in 250 frame blocks, with 1 s uniform gray screen between blocks. The total duration of each model simulation was 89 s, which included 18 individual white noise stimulus blocks. All simulations used a time resolution of 0.1 ms.

Each LGN spike was convolved with an excitatory post-synaptic conductance (EPSG). The EPSG consisted of two exponentials, one for the rise (τ = 0.5 ms), and one for the decay, (τ = 2 ms). This produced an EPSG that rose and decayed faster than the average EPSPs observed in vivo (*Sedigh-Sarvestani et al., 2017*). The faster dynamics of the EPSG were then smoothed by the model cell's membrane, giving rise to average EPSPs similar to those measured in vivo. The EPSG was 'injected' into the target V1 cell producing Vm depolarizations relative to a resting Vm of −65 mV. When the Vm reached the spike-threshold of −55 mV, an action potential was fired and the Vm was reset to −60 mV. A 2 ms refractory period was observed for cortical spiking. The cortical cell membrane properties were based on our in vivo measurements for L4 RS cells with an Rm of 47 MOhms, and time-constant of 6 ms.

We generated and compared two versions of the model, one with up to 50 'reliable' thalamocortical synapses, and the second with up to 50 'unreliable' synapses. Reliable synapses converted every pre-synaptic LGN spike into a EPSG with constant amplitude (*Figure 6C*, left). Unreliable synapses included a variable proportion of failed spikes, which were not convolved with EPSGs, and their amplitude was not constant but varied between values which were larger and smaller than the mean EPSP observed in our data (*Figure 6C*, right). For a specific thalamocortical synapse, the average amplitude of EPSPs produced by the reliable synapses was equal to the average amplitude of EPSPs in the corresponding unreliable synapse (*Figure 6C*, bottom). This ensured that the total input current for reliable and unreliable synapses was equal, allowing for a controlled comparison of the effect of thalamocortical transmission failures and variability on V1. Unless otherwise specified, spike failures in the unreliable model were randomly selected for each LGN cell's spike-train following the proportions observed in the data (*Figure 6D*).

Note that all synapses did not produce the same size average EPSP rather the average EPSP amplitude was selected from the distribution of observed average EPSPs in our dataset, with a normalization factor of 6 × 108 to convert amplitude (mV) to conductance (nS), so that the distribution of EPSPs produced by the model was similar to the distribution measured in vivo. The average EPSG amplitude across all 50 inputs was 0.89 nS.

For unreliable synapses, the %failed spikes and EPSP amplitude variance corresponding to the average EPSP amplitude were selected from our measured dataset. Therefore in the model, as in our measured dataset, synapses with larger average EPSP amplitudes had lower failure rates and EPSP amplitude variance.

Each triplet of synaptic properties (average, variance of EPSP amplitude and failure rates) was randomly assigned to a pre-synaptic LGN cell. We simulated the unreliable and reliable versions of the model n = 10 times, with each trial having a different random assignment of synaptic parameters.

Additional model details, including parameters used, can be found in Appendix 1.

## Acknowledgements

We thank Morgan Taylor for critical feedback on the manuscript, and Ivan Fernandez-Lamo and Leif Vigeland for their help with experiments.

# Additional information

## Funding

| Funder | Grant reference number | Author |
|---|---|---|
| National Eye Institute | R01EY027205 | Larry A Palmer<br>Diego Contreras |
| National Institutes of Health | F32EY026463 | Madineh Sedigh-Sarvestani |

The funders had no role in study design, data collection and interpretation, or the decision to submit the work for publication.

## Author contributions

Madineh Sedigh-Sarvestani, Conceptualization, Data curation, Software, Formal analysis, Validation, Investigation, Visualization, Methodology, Writing—original draft, Writing—review and editing; Larry A Palmer, Conceptualization, Data curation, Formal analysis, Supervision, Funding acquisition, Investigation, Methodology, Project administration, Writing—review and editing; Diego Contreras, Conceptualization, Data curation, Supervision, Funding acquisition, Project administration

## Author ORCIDs

Madineh Sedigh-Sarvestani (ID) http://orcid.org/0000-0002-8735-2927
Diego Contreras (ID) http://orcid.org/0000-0003-0197-9882

## Ethics

Animal experimentation: This study was performed in strict accordance with the recommendations in the Guide for the Care and Use of Laboratory Animals of the National Institutes of Health. All of the animals were handled according to approved institutional animal care and use committee (IACUC) of the University of Pennsylvania (Protocol # 803477). All surgery was performed under sodium pentobarbital or propofol anesthesia, and every effort was made to minimize suffering.

## Decision letter and Author response

Decision letter https://doi.org/10.7554/eLife.41925.024
Author response https://doi.org/10.7554/eLife.41925.025

# Additional files

## Supplementary files

• Transparent reporting form
DOI: https://doi.org/10.7554/eLife.41925.019

## Data availability

All data generated or analysed during this study are included in the manuscript and supporting files. Raw data and MATLAB code have been uploaded to Dryad (http://dx.doi.org/10.5061/dryad.57pv818).

The following dataset was generated:

| Author(s) | Year | Dataset title | Dataset URL | Database and Identifier |
|---|---|---|---|---|
| Sedigh-Sarvestani M, Palmer L | 2019 | Data from: Thalamocortical synapses in the cat visual system in vivo are weak and unreliable | http://dx.doi.org/10.5061/dryad.57pv818 | Dryad Digital Repository, 10.5061/dryad.57pv818 |

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

## Appendix 1

DOI: https://doi.org/10.7554/eLife.41925.020

### Evaluating cluster quality

Clusters were evaluated and included in the dataset using a combination of factors. Although we report the commonly used Lratio and Isolation Distance metrics (*Figure 1—figure supplement 1*), we did not use a threshold on either value to include or reject clusters. In part, due to lack of consensus in the literature and in part because in certain cases, these metrics did not support our visual assessment of cluster quality. We therefore relied on manual cluster-cutting and visual verification and refinement.

Although we used software that had automated cluster cutting capabilities, we manually cut clusters by visualizing the cluster space across many dimensions (features of action potentials), and visually verified each cluster included in our data. In doing so, we used several criteria for cluster definition:

1. The shape and distance of the cluster relative to the 'noise' cluster: We only chose clusters that were visually separate from the noise cluster.
2. The distribution of inter-spike intervals: We excluded any traces with an ISI <1 ms.
3. The shape of action potentials included in the cluster: We manually cut action potentials that did not fit the average shape.
4. The receptive field of the cluster: We excluded any clusters that did not produce a clear center-surround receptive field characteristic of LGN cells.

Having viewed and assessed each single cluster as detailed above, we are confident in the quality of our clusters. We saw no relationship between the magnitude of EPSPs, or the % detectability, and either the Lratio or Isolation Distance cluster quality metrics.

### Validating single-spike EPSP detector

As a secondary and independent verification of single-spike EPSP categorizations (*Figure 2E*, *Figure 3*), we used a linear discriminant based classifier to independently categorize all single-spike traces from all connected pairs and compared the output of the two methods. The features used for the classification were the mean, max, and standard deviation of the single-spike EPSP trace as well as the mean and max of the first derivative of the single-spike EPSP trace. For each connected pair, we hand-labeled a randomly selected 10% of single-spike traces and used this to train the classifier (*Figure 3—figure supplement 1A*), which then categorized the remaining 90% of single-spike traces into either 'detectable' or 'undetectable' classes. For each single-spike EPSP, the classifier produced a probability that the trace included a detectable EPSP. This probability was thresholded at 0.5 to produce the final class labels.

Across all connected pairs, the categorization from the two detectors matched an average of $83 \pm 3.8\%$, and this increased to $87 \pm 2.9\%$ when the training set included 25% of single-spike traces. Both methods classified a similar number of undetectable EPSPs (*Figure 3—figure supplement 1C*): A population average of $37 \pm 12\%$ for the automated detector and $35 \pm 15\%$ for the classifier. Misclassifications (black dots, *Figure 3—figure supplement 1A*) were most often false-positives, single-spike traces without a sharp depolarization that were labeled as single-spike EPSPs. Although both methods produced similar categorizations, the detector had fewer false positives as revealed by visual inspection. We therefore used categorizations produced by the detector.

### Role of instantaneous Vm and ISI on synaptic reliability and variability

For each pre-synaptic spike, we measured (1) the average Vm ±1 ms around each LGN spike (Vm(tspike), *Figure 4—figure supplement 1A*) and (2) the inter-spike-interval (ISI), or the time since the previous spike from the same pre-synaptic cell. We were interested in the role of Vm

since the constant fluctuations of Vm in vivo, either spontaneous or in response to visual stimulation, likely change the driving force of the single-spike EPSP, potentially leading to changes in its amplitude. We were interested in the role of spiking history quantified as ISI, since many central synapses are known to undergo spike-history dependent short-term plasticity (*Regehr, 2012*).

We looked for a correlation between single-spike trace amplitudes (amp, *Figure 3A*) and Vm(tspike) and ISI. Across our population of 36 connected pairs, 16 exhibited a significant correlation between amp and Vm(tspike), with an average correlation of $-0.13 \pm 0.056$, consistent with a reduction in driving force during depolarization. In contrast, 10 of 36 cell-pairs exhibited a significant correlation between amp and ISI, with an average of $0.11 \pm 0.10$, suggesting an average increase of amplitude with longer ISI indicative of short-term depression.

To understand how much of the variability in single-spike amplitudes could be explained by Vm(tspike) and ISI, we fit our data to a simple multiple linear regression model (*Figure 4—figure supplement 1B*). We found that the majority of cell-pairs (28/36) had non-significant regression coefficients either for Vm(tspike) or ISI (*Figure 4—figure supplement 1B*, open circles), consistent with the majority of cell-pairs exhibiting non-significant correlation coefficients. For cell-pairs with significant correlation coefficients for both Vm(tspike) and ISI (*Figure 4—figure supplement 1B*, filled circles), the coefficient for Vm(tspike) was of similar magnitude to that for ISI. We note that although all significant regression coefficients for Vm(tspike) were negative, the significant coefficients for ISI were both positive and negative, suggesting the presence of both short-term depression and facilitation. We conclude that for the majority of cell-pairs, the instantaneous Vm and ISI are not significantly correlated with single-spike trace amplitude, likely because the data are too noisy at single-spike resolution.

To average out some of the noise at the single-spike level, we divided the Vm and ISI for each cell-pair into 1 st (0–25%) and 3rd (50–75%) quartiles (*Figure 4—figure supplement 1C, D*). We then compared the properties of the single-spike trace amplitude distributions for the top and bottom Vm and ISI quartiles for each cell-pair. Across the population of 36 cell-pairs, the average amp during periods of low Vm(tspike) ($0.57 \pm 0.28$ mV) was larger than that during more depolarized high Vm(tspike) ($0.41 \pm 0.20$ mV), but this difference did not pass significance (two sample KS test, p=0.055). The average %undetectable during low Vm(tspike) was significantly smaller than that during more depolarized high Vm(tspike) ($32 \pm 12$% vs $42 \pm 12$%, p=0.028), and consistent with this, the average CV during low Vm(tspike) was also significantly smaller than that during more depolarized Vm ($0.85 \pm 0.45$ vs. $2.0 \pm 1.8$). These trends (*Figure 4—figure supplement 1D*) suggest that single-spike evoked EPSPs are, on average, larger and less variable during hyperpolarized Vms. However, the correlation between Vm and EPSP characteristics are small, and not detectable with single-spike resolution in all cell-pairs (*Figure 4—figure supplement 1B*).

In evaluating the effect of short (1 st quartile) and long (3rd quartile) ISIs on EPSP characteristics, the trends were consistent with short-term depression, but none passed significance (*Figure 4—figure supplement 1F*). Across the population, the average amp for spikes with short ISI was comparable to that for long ISIs ($0.43 \pm 0.18$ Vm vs. $0.57 \pm 0.34$ Vm, p=0.18), although the trend was in the direction of short-term depression. The average % undetectable for spikes with short ISI was $40 \pm 11$%, smaller but not significantly different from that for spikes with longer ISIs ($34 \pm 14$%, p=0.055). The population average CV for spikes with short and long ISIs also did not pass significance ($1.3 \pm 0.54$ vs $1.1 \pm 0.57$, p=0.46). All together, the population data shows a small and statically insignificant effect of ISI on EPSP dynamics. However, we wondered whether the presence of both short-term depression and short-term facilitation, across different connected pairs, in the population reduced the detectable effect of ISI on EPSP dynamics. Therefore, as detailed in the main text, we analyzed short-term synaptic plasticity for each individual cell-pair using classic methodology (*Figure 5*).

## Model of L4 V1 neuron

We modeled a typical L4 V1 neuron using a leaky integrate-and-fire (IF) model, driven by excitatory LGN input and optionally, balanced excitatory and inhibitory synaptic background activity.

$$C_m \frac{dV}{dt} = -\left[ \frac{1}{R_m}(V - E_L) + (I_{syn-LGN} + I_{syn-bkgrnd}) \right] \tag{2}$$

The model neuron had a resting potential of −65 mV, an input resistance of 47 MOhm and a membrane time-constant ($\tau_m$) of 6 ms, typical values for our dataset of intracellular recordings with sharp electrodes (**Cardin et al., 2007**). When the Vm reached −55 mV, the neuron fired a 55 mV action potential and was reset to −65 mV. A spike refractory period of 2 ms was imposed. The model was driven by spikes from a group of up to 50 LGN neurons, recorded in the anesthetized cat during the presentation of white noise stimuli. Each pre-synaptic spike LGN produced an excitatory post-synaptic conductance (EPSG) in the V1 cell.

Each single-spike EPSG waveform triggered by a pre-synaptic spike was produced according to the function below (5). A synaptic latency of 2 ms was imposed before the onset of the EPSC. τrise of 0.5 ms and a τdecay of 2 was used for all synapses. For each LGN neuron, Gpeak was set proportional to a randomly selected value from our database of mean EPSP amplitudes. The proportionally constant was used so that the mean evoked EPSP fell within the range of our experimental observations (~0.5 mV).

$$EPSG(t) = G_{peak} \left( 1 - \frac{t}{e^{\tau} rise} \right)^5 \frac{t}{e^{\tau} decay} \tag{3}$$

Optionally, short-term depression at each synapse was imposed by applying a gain factor to each single-spike EPSG based on the time since the previous pre-synaptic spike. An exponential function, fit to the normalized EPSP amplitude vs. ISI curve from a sample synapse in our dataset (**Figure 5A**), was used to calculate the depression gain factor applied to each single-spike EPSC:

$$STDgain = max(0.3, c + aISI^b) \tag{4}$$

Where c is 206, a is 206, and b is 5.5e-4. The maximum function ensures that the STD gain does not degenerate to low values for small ISIs.

The synaptic current due to LGN spike input is then described by:

$$I_{syn-LGN}(t) = g_{syn-LGN}(t)(V - E_e) \tag{5}$$

where $g_{syn-LGN}(t)$ is the conductance time-series, calculated by convolving the combined input spike train from all cells with their corresponding single-spike EPSG(t)*STDgain. $E_e$ is the reversal potential of glutamate: 0 mV.

L4 V1 is characterized by dense synaptic connections from excitatory and inhibitory neurons, only ~5–10% of which can be contributed to excitatory thalamic afferents, which give rise to a high conductance state characterized by fluctuating Vm and irregular firing (**Destexhe et al., 2001**). The background synaptic activity giving rise to this high conductance state was simulated by injecting excitatory and inhibitory conductances, according to previously published methods (**Destexhe et al., 2001**).

$$I_{syn-bkgrnd}(t) = g_e(t)(V - E_e) + g_i(t)(V - E_i) \tag{6}$$

where Ei=-75 mV. The conductances were described by a stochastic process, with the following update rules:

$$g_{e/i}(t+dt) = g_{e/io} + \left[g_{e/i}(t) - g_{e/io}\right]e^{\frac{-dt}{\tau_{e/i}}} + A_{e/i}\,N_1(0,1)$$

$$A_{e/i} = \sqrt{\sigma_{e/i^2}\left[1 - exp(\tfrac{-2dt}{\tau_{e/i}})\right]}$$

(7)

where $g_{e/i0}$ are the mean background excitatory and inhibitory conductance, $\tau_{e/i}$, are time constants of excitatory and inhibitory synapses, N1 (0,1) is a zero-mean unit variance Gaussian process, and $\sigma_{e/i}$ is the standard deviation of excitatory and inhibitory conductances. The parameters were set so that a model with only background synaptic activity would produce membrane potential fluctuations and firing rates similar to those in our measured dataset of intracellular L4 V1 neurons responding to a gray screen. However, we note that these parameters produced firing rates in response to visual stimulation (or input LGN spikes) that were higher than firing rates observed in vivo during the response to visual stimulation (compare *Figure 7* and *Figure 7—figure supplement 3*).

$g_{e0,i0}$: [2 nS, 8 nS]
$\sigma_{e,i}$: [1 nS, 3 nS]
$\tau_{e,i}$: [8 ms, 9 ms]

## Testing the effect of pre-synaptic spike failure distributions on post-synaptic firing

In the unreliable version of the model used in the main text, we randomly selected which pre-synaptic LGN spikes would fail to produce a single-spike EPSP in the post-synaptic model cell. However, we wondered whether spike-failures that were correlated across the pre-synaptic population may change our findings that unreliable synapses produce enhanced post-synaptic firing. Both random and correlated failure distributions are biologically plausible, with correlated failures arising from post-synaptic sources of failure such as shunting inhibition and uncorrelated random failures arising from pre-synaptic sources. Our dataset includes 5 V1 cells with three connected LGN cells. We found no evidence of correlated failures across these 3 LGN cells connected onto the same V1 cell. However, in 4 of the five cases the LGN receptive fields did not exhibit significant overlap, thus precluding a meaningful analysis of correlated firing and therefore correlated failures. Thus, we use a modeling approach to manipulate the distribution of failures and their impact on population synchrony.

To test the effect of the distribution of failures across synapses, we added a model with a second type of unreliable synapses, wherein undetectable EPSPs ('failures') were correlated across the population of synapses (*Figure 7—figure supplement 2A*; blue dots). The two types of unreliable synapses have different distributions of failures across LGN cells, but the failure rate and the overall conductance input for all synapses is matched, as evidenced by comparable average Vm across all models (*Figure 7—figure supplement 2B*, left).

*Figure 7—figure supplement 2A* shows a sample simulation of the LGN inputs and resulting V1 response for a model. The V1 RF has such a low signal-to-noise ratio that it is indistinguishable from the background. This suggests that synchronous spikes are crucial for information transfer in the thalamocortical circuit. The effect of failures and synaptic variability on information transmission is beyond the scope of this paper, but a recent simulation study suggests that unreliable synapses enhance the transfer of certain types of fast signals (*Gatys et al., 2015*). For slower signals, unreliable synapses likely reduce information transfer, depending on the rate and distribution of failures across synapses and the population synchrony (*Zador, 1998*).

Despite the lack of a V1 RF produced by unreliable synapses with correlated failures, we found that this model produced increased firing in V1 (*Figure 7—figure supplement 2D*, blue traces) relative to the reliable model (*Figure 7—figure supplement 2D*, black traces). As before, this increased firing was due to an increase in Vm SD produced by unreliable synapses, and the exponential relationship between Vm SD and firing rate (*Figure 7—figure supplement 2C*). There was no significant difference between cortical firing produced by unreliable synapses with random failures (*Figure 7—figure supplement 2D*, red traces), and unreliable synapses with correlated failures (blue traces). This suggests that unreliable

synapses enhance V1 firing regardless of the distribution of spike failures or undetectable EPSPs.

