## [Decision Letter]

Thank you for submitting your article "Thalamocortical synapses in the cat visual system in vivo are weak and unreliable" for consideration by *eLife*. Your article has been reviewed by Ronald Calabrese as the Senior Editor, a Reviewing Editor, and three reviewers. The reviewers have opted to remain anonymous.

The reviewers have discussed the reviews with one another and the Reviewing Editor has drafted this decision to help you prepare a revised submission.

Summary:

These data are hard-won, from a highly challenging preparation, with important implications for models of thalamocortical function. There have been very few recordings of connected thalamocortical pairs in vivo, and even fewer with intracellular recordings. Using this approach, the authors report that the LGN-V1 synapse is highly unreliable in vivo, in contradiction to previous slice studies. They build a model of a L4 neuron receiving LGN connections, which predicts that unreliability increases cortical firing rates. There are only a handful of similar datasets to this one and absolutely no others in visual cortex; these are truly unique data. The analyses are clever and original. Much of what we base our general ideas of thalamocortical interactions on is thus based on in vitro, rather than in vivo, data. Given the authors' careful quantitative approach, the current findings are thus likely to have a substantial impact in the field.

Essential revisions:

There are a few major elements of the manuscript that could be substantially improved through text clarification and there are several points that could benefit from additional analysis or enhanced discussion (listed as the first major points below). No additional experiments seem necessary.

1) The authors should acknowledge that they cannot measure synaptic reliability in vivo. This does not detract from the impact of the findings, but the use of 'reliability' is misleading in the broader context of the field. Reliability is the term used to describe the variance in synaptic transmission events-release of neurotransmitter that successfully binds to receptors and causes a synaptic current flux. It is not currently possible to measure this in vivo. What the authors are measuring is the functional impact of the synaptic event on the postsynaptic neuron's membrane potential. This is an important measure, as it is directly linked to postsynaptic spike output, but it is distinct from the technical term 'reliability'. The authors cannot distinguish in vivo between (1) synaptic failures-which do impact reliability estimates-and (2) moments where a successful synaptic event is lost to noise, shunting, etc. – which do not impact synaptic reliability but do affect functional postsynaptic impact.

2) The model makes an interesting prediction (unreliable synapses are helpful), but the authors could do a better job of making this theoretical case. First, they compare the predicted enhancement of cortical discharges given unreliability versus thalamic synchrony, which is interesting. However, the synchrony mechanism requires recurrent cortical excitation and feedforward inhibition, which a few papers cited by the authors (Wang et al., 2010; Bruno, 2011) make a strong point of and which are well documented experimentally. The model here, however, lacks both recurrent excitation and feedforward inhibition, so it is hard to know how unreliable synapses and thalamic synchrony compare or interact. Second, these two elements could lessen the effect of unreliable synapses. Since the model is the only evidence that unreliable synapses are feature not flaw, the authors should consider an expanded simulation.

3) The authors must include in the Discussion section a mention of the caveats associated with the use of anesthesia, especially as the data represent a mix of thiopental and propofol preparations. Even light anesthesia alters the statistics of thalamic firing, and almost no cortical recordings have ever been made under propofol anesthesia.

4) The discussion in subsection “Detecting Single-spike EPSPs” is not clear. For a multisite EPSP (i.e. one arising from multiple synaptic contacts from a single thalamic axon onto a single cortical neuron) to fractionate, i.e. be resolved as separate EPSPs they would likely need to be distributed at very distinct electrotonic distances from the presumed somatic recording. In addition, I think this discussion is not particularly relevant to the interpretation of the results. The bottom line is that it would be very difficult to resolve whether an individual paired response results from a single synapse or a group of synapses, whether they are clustered or not. The large amplitude variability of responses (subsection “Validating single-spike EPSP detector”) is in fact consistent with either multivesicular release at individual synapses, or multi-synapse release, with a possible binomial distribution (subsection “Thalamocortical synapses are unreliable and produce highly variable EPSPs”) of amplitudes, as expected from standard synaptic models. This needs to be better explained.

5) While both the White Noise and Grating stimuli produce similar results, there seems to be a significant difference, which the authors gloss over. Figure 4H suggests that synapses are stronger and more unreliable during Grating stimulation than during White Noise stimulation. The authors should report the results of a paired test and discuss appropriately.

6) KS tests, which evaluate the whole distribution, are overly sensitive in that the slightest difference-even of a tail-produces a significant result. Unless the authors wish to focus on potential subpopulations (tail effects) in their analysis, then the results around short vs long ISIs (Figure 5C,D) would be more compelling if the authors could construct a simple analysis testing the median or means of these data, which appears to be their real question.

7) The present results seem somewhat more consonant with previous findings than the authors suggest. For instance, the authors did not find a correlation of depression and pre-synaptic firing rate, which they say is at odds with Boudreau and Ferster 2005's study of thalamocortical depression (subsection “Thalamocortical synapses exhibit short-term synaptic plasticity”). Might it not be that the present study is simply at the high end of pre-synaptic firing rates (B&F's at the lower end) and the depression is largely maxed out? The authors could explain better the extent to which this is or is not a discrepancy. A second instance is where the authors say that their EPSP amplitudes are modest even when compared to the in vivo measurements in the somatosensory system (subsection “Thalamocortical strength, reliability, and variability in vivo”). Bruno and Sakmann, 2006 reported similar average EPSPs as the 'all single-spikes' in Figure 4 here, and Schoonover et al., 2014 claim similar strength of unitary EPSPs (failures removed) as the 'detectable EPSPs' in Figure 4 here. This is a positive not a negative of the manuscript because it suggests that different sensory systems might be exploiting similar mechanisms and principles. A comment about the authors' discovery may be merited.

8) Simply reporting that spikes were manually clustered is not sufficient. The authors should report% spikes occurring with 1 ms of each other along with cluster isolation metrics.

9) Subsection “Effect of thalamocortical variability on L4”: 'Data not shown' is not really acceptable, the authors should include simulations in supplementary data or not mention them at all.

10) Subsection “Thalamocortical synapses are unreliable and produce highly variable EPSPs”. IPSPs do not need to be hyperpolarizing to influence membrane responses and especially EPSPs. The associated in increase in conductance will itself decrease amplitude of EPSPs, all other things being equal. Overall, I would like to see a discussion of the influence of dynamics of membrane conductance in terms of the influence of EPSPs on Vm and on their amplitude and detectability.

11) Previous in vitro and in vivo studies have focused on layer IV itself, while this study more broadly samples deeper cortical layers. This may influence some of the results. In particular, it has been reported that thalamocortical inputs onto FS cells in layer IV tend to be stronger and more reliable than the inputs onto RS cells. The relative similarity in EPSPs reported here may result in part from undersampling layer IV.

[Editors' note: further revisions were requested prior to acceptance, as described below.]

Thank you for resubmitting your work entitled "Thalamocortical synapses in the cat visual system in vivo are weak and unreliable" for further consideration at *eLife*. Your revised article has been favorably evaluated by Ronald Calabrese (Senior Editor), a Reviewing Editor, and two reviewers.

The manuscript has been improved but there are some remaining issues that need to be addressed before acceptance, as outlined below in the next section. The reviewing editor will make the final assessment of the revised manuscript.

Essential revisions:

Overall, the authors have addressed most previous concerns. The text is also generally more readable and clear. The manuscript now focuses on the dynamics of the LGN-V1 synapse in vivo and simply advocates that this synaptic variability enhances post-synaptic fluctuations and therefore firing rates. While a more extensive model that addressed the intriguing issue of synchrony could provide substantial insight, we agree with the authors that a full examination deserves a separate study.

- There does not seem to be a clear set of criteria for unit inclusion based on cluster metrics. The plot in Figure 1—figure supplement 1 does not make it easy to evaluate the isolation distances, but some points appear to be very low.

- The final paragraph of the Introduction is somewhat confusing as written. Perhaps the authors meant "high variability of the thalamocortical synapse, rather than being irrelevant system noise, does play a functional role…"?

- There are a few places where text changes have not been fully incorporated and should be fixed, e.g. subsection “Thalamocortical synapses are unreliable and generate highly variable EPSPs”, Discussion section, figure legend for Figure 1—Figure supplement 1.

- The authors are perhaps overly modest in citing their own previous work. The Contreras group has published several other studies (2007, 2010) on the synaptic and intrinsic properties of RS and FS cells in cat visual cortex that are relevant to the current findings.

- Subsection “Thalamocortical synapses are unreliable and generate highly variable EPSPs”, "…there is a population of very large (>4 mV) EPSPs triggered by LGN input." This unusual population is interesting and warrant comparison with the more typical 0.5-1.0 mV EPSPs. Most of the traces shown are the common small type. Figure 3B green appears to show a larger one that is 2-3 mV at most. These very large EPSPs could result from nonlinear voltage-dependent mechanisms that might have a different shape from normal EPSPs, bursts of LGN spikes, etc. The green traces indeed exhibit a long shoulder suggestive of voltage-dependent mechanisms. Peak normalizing the average small and large EPSPs should allow visual comparison of their shapes.

- On a related note, subsection “Detecting Single-spike EPSPs” states that small "undetectable" EPSPs are unlikely to influence postsynaptic Vm. While I agree with this statement generally, I think the authors have to qualify it with "at least under our conditions". One can easily imagine nonlinear voltage-gated channels that make such small EPSPs potent under different conditions (e.g., state dependent neuromodulation of those channels, disinhibitory circuits, etc.). Also, the interim results summary paragraph, subsection “subsection “Thalamocortical synapses are unreliable and generate highly variable EPSPs”, is confusing. Consider "…including a large percentage of EPSPs that do not visibly affect somatic Vm and are unlikely to contribute to the cortical output." As noted in point 2, this is not guaranteed. More so it doesn't seem to allow for the possibility of synaptic failures. This and the surrounding text could be read to mean they "undetected" EPSPs are always small and never failures.

The authors have improved the short-term depression section, but there is still some difficulty following the logic of the predictions (subsection “Short-term synaptic plasticity in vivo”). Boudreau and Ferster, 2005 electrically stimulated LGN inputs in anesthetized cats and predicted that depression would be near saturated in the awake animal. The present study, also under anesthesia, evoked LGN spikes by visual stimulation and measured the synaptic input. They find that LGN firing rate does not correlate with the amount of depression (short vs long ISI) they observe, which in any case is about a 17% decrease. However, Boudreau and Ferster's depression curve does show a slight similar decrease from the first to later stimuli. Additionally, the present study only has 3 data points above 10 Hz-how can it be used to make claims about what happens for awake firing rates? 1-10 Hz is already in the range of awake spontaneous activity.

---

## [Author Response]

Summary:These data are hard-won, from a highly challenging preparation, with important implications for models of thalamocortical function. There have been very few recordings of connected thalamocortical pairs in vivo, and even fewer with intracellular recordings. Using this approach, the authors report that the LGN-V1 synapse is highly unreliable in vivo, in contradiction to previous slice studies. They build a model of a L4 neuron receiving LGN connections, which predicts that unreliability increases cortical firing rates. There are only a handful of similar datasets to this one and absolutely no others in visual cortex; these are truly unique data. The analyses are clever and original. Much of what we base our general ideas of thalamocortical interactions on is thus based on in vitro, rather than in vivo, data. Given the authors' careful quantitative approach, the current findings are thus likely to have a substantial impact in the field.Essential revisions:There are a few major elements of the manuscript that could be substantially improved through text clarification and there are several points that could benefit from additional analysis or enhanced discussion (listed as the first major points below). No additional experiments seem necessary.1) The authors should acknowledge that they cannot measure synaptic reliability in vivo. This does not detract from the impact of the findings, but the use of 'reliability' is misleading in the broader context of the field. Reliability is the term used to describe the variance in synaptic transmission events-release of neurotransmitter that successfully binds to receptors and causes a synaptic current flux. It is not currently possible to measure this in vivo. What the authors are measuring is the functional impact of the synaptic event on the postsynaptic neuron's membrane potential. This is an important measure, as it is directly linked to postsynaptic spike output, but it is distinct from the technical term 'reliability'. The authors cannot distinguish in vivo between (1) synaptic failures-which do impact reliability estimates-and (2) moments where a successful synaptic event is lost to noise, shunting, etc. – which do not impact synaptic reliability but do affect functional postsynaptic impact.

We thank the reviewers for raising this very important point. We have added a clarifying statement in the Results section in conjunction to Figure 2A indicating the limitations in vivo mentioned by the reviewer and clearly explaining how we use this term. We have also included a clear explanation in Discussion section. We make it clear and acknowledge that we can’t measure true synaptic reliability in vivo. Here we use the term reliability to simply indicate the% of times that single LGN spikes generate detectable EPSPs in the soma of our recorded neurons. We have also removed many instances of the term reliability and substituted the term by referring to the% of undetectable EPSPs, which is really what we measure.

2) The model makes an interesting prediction (unreliable synapses are helpful), but the authors could do a better job of making this theoretical case. First, they compare the predicted enhancement of cortical discharges given unreliability versus thalamic synchrony, which is interesting. However, the synchrony mechanism requires recurrent cortical excitation and feedforward inhibition, which a few papers cited by the authors (Wang et al., 2010; Bruno, 2011) make a strong point of and which are well documented experimentally. The model here, however, lacks both recurrent excitation and feedforward inhibition, so it is hard to know how unreliable synapses and thalamic synchrony compare or interact. Second, these two elements could lessen the effect of unreliable synapses. Since the model is the only evidence that unreliable synapses are feature not flaw, the authors should consider an expanded simulation.

We fully agree with the caveats raised by the reviewer concerning the need to add recurrent excitation and feedforward inhibitory connections to the thalamocortical model in order to test the effects of variability and thalamocortical synchrony. However, adding feedforward inhibition to the model in a useful way would require a huge number of assumptions about feedforward inhibition and it would take the model far away from ‘biologically constrained’. We searched for published thalamocortical models which could more easily incorporate cortical network connectivity in order to properly evaluate the interaction of synchrony and variability, but we found nothing with the particular set of network constrains required for our purpose. In summary, to add network parameters to our thalamocortical model would require to vary the parameters (strength, similarity of tuning, timing, etc.) and see how each parameter set impacts the effect of synchrony on cortical firing. This is a complex multidimensional parameter space which would require a study on its own. We believe this would be a huge distraction from the experimental data in the paper, which took us over 5 years to collect, so we have decided to entirely remove the synchrony portion from the manuscript and leave it for a future study centered on computational modeling.

However, we chose the leave the portions of the supplementary figure that dealt with correlated (as opposed to random) synaptic failures (Figure 7—figure supplement 2) because we felt that synchrony aside, it was important to show that our simulation results were not contingent on the particular distribution of failed LGN spikes.

3) The authors must include in the Discussion section a mention of the caveats associated with the use of anesthesia, especially as the data represent a mix of thiopental and propofol preparations. Even light anesthesia alters the statistics of thalamic firing, and almost no cortical recordings have ever been made under propofol anesthesia.

We agree with the reviewer and have added appropriate comments in the Discussion section alerting the reader to the problems of anesthesia.

4) The discussion in subsection “Detecting Single-spike EPSPs” is not clear. For a multisite EPSP (i.e. one arising from multiple synaptic contacts from a single thalamic axon onto a single cortical neuron) to fractionate, i.e. be resolved as separate EPSPs they would likely need to be distributed at very distinct electrotonic distances from the presumed somatic recording. In addition, I think this discussion is not particularly relevant to the interpretation of the results. The bottom line is that it would be very difficult to resolve whether an individual paired response results from a single synapse or a group of synapses, whether they are clustered or not. The large amplitude variability of responses (subsection “Validating single-spike EPSP detector”) is in fact consistent with either multivesicular release at individual synapses, or multi-synapse release, with a possible binomial distribution (subsection “Thalamocortical synapses are unreliable and produce highly variable EPSPs”) of amplitudes, as expected from standard synaptic models. This needs to be better explained.

We agree with the reviewer that this is confusing, we decided to entirely remove this discussion, it is not entirely relevant in vivo (as the reviewer says) and highly distracting from the main points of the manuscript.

5) While both the White Noise and Grating stimuli produce similar results, there seems to be a significant difference, which the authors gloss over. Figure 4H suggests that synapses are stronger and more unreliable during Grating stimulation than during White Noise stimulation. The authors should report the results of a paired test and discuss appropriately.

The data in Figure 4H suggests that during grating stimulation, the synapses are weaker (produce smaller EPSPs) and more unreliable. We ran paired t-tests to compare the response magnitude, variability, and LGN firing rate associated with the same synapse during presentation of white noise or grating stimuli and found the following p values:

Mean EPSP-amplitude: 0.0055

Standard deviation of EPSP-amplitude: 0.070

CV: 0.71

%undetectable: 0.075

LGN firing rate: 0.16

Therefore, it appears that during white noise stimuli, thalamocortical synapses produce stronger and somewhat more detectable EPSPs, though the differences are small. One possible explanation may be an increased coordination of thalamic inputs during grating stimuli, producing greater feedforward inhibition which results in decrease amplitude and reliability of EPSPs.

We thank the reviewer for bringing this to our attention and have added comments in the text.

6) KS tests, which evaluate the whole distribution, are overly sensitive in that the slightest difference-even of a tail-produces a significant result. Unless the authors wish to focus on potential subpopulations (tail effects) in their analysis, then the results around short vs long ISIs (Figure 5C,D) would be more compelling if the authors could construct a simple analysis testing the median or means of these data, which appears to be their real question.

We thank the reviewer for this important correction. We used a two-sample t-test to test the null hypothesis that the distributions in Figure 5C and 5D had the same mean. We confirmed that the distributions were approximately Gaussian, thus making the t-test appropriate. Our results remain the same. Namely, the distributions in Figure 5C and 5D come from populations with different means. We have updated the text and the figure with results of the t-test. Figure 5C: p=1.51e-6, 5D: p=0.0019.

7) The present results seem somewhat more consonant with previous findings than the authors suggest. For instance, the authors did not find a correlation of depression and pre-synaptic firing rate, which they say is at odds with Boudreau and Ferster 2005's study of thalamocortical depression (subsection “Thalamocortical synapses exhibit short-term synaptic plasticity”). Might it not be that the present study is simply at the high end of pre-synaptic firing rates (B and F's at the lower end) and the depression is largely maxed out? The authors could explain better the extent to which this is or is not a discrepancy. A second instance is where the authors say that their EPSP amplitudes are modest even when compared to the in vivo measurements in the somatosensory system (subsection “Thalamocortical strength, reliability, and variability in vivo”). Bruno and Sakmann, 2006 reported similar average EPSPs as the 'all single-spikes' in Figure 4 here, and Schoonover et al., 2014 claim similar strength of unitary EPSPs (failures removed) as the 'detectable EPSPs' in Figure 4 here. This is a positive not a negative of the manuscript because it suggests that different sensory systems might be exploiting similar mechanisms and principles. A comment about the authors' discovery may be merited.

We have rewritten our comparison with the studies of Boudreau and Ferster. Our numbers match theirs quite well, but we found that STPD magnitude is independent of firing rate within our range of 2 to 24 Hz, if such rates are observed in awake state in cats then there is a possibility that synaptic depression would still be observed in the waking state. Indeed, as the reviewer points out, our numbers are similar to Randy’s, both from Germany (0.9 mV) and New York (1.0 mV).

8) Simply reporting that spikes were manually clustered is not sufficient. The authors should report% spikes occurring with 1 ms of each other along with cluster isolation metrics.

We have included cluster metrics in Figure 1—figure supplement 1.

9) Subsection “Effect of thalamocortical variability on L4”: 'Data not shown' is not really acceptable, the authors should include simulations in supplementary data or not mention them at all.

We have added a supplementary figure (Figure 7—figure supplement 3) that includes this data, as well additional related model details in the Supplementary materials.

10) Subsection “Thalamocortical synapses are unreliable and produce highly variable EPSPs”. IPSPs do not need to be hyperpolarizing to influence membrane responses and especially EPSPs. The associated in increase in conductance will itself decrease amplitude of EPSPs, all other things being equal. Overall, I would like to see a discussion of the influence of dynamics of membrane conductance in terms of the influence of EPSPs on Vm and on their amplitude and detectability.

We were only referring to the fact that the average of undetectable EPSPs is slightly negative and not simply zero like the average of shuffle traces. We conjectured, that such negative potential could be due to a preceding EPSP that makes the “baseline” positive with respect to the point where we measure amplitude, or to the presence of a true IPSP. The reviewer is referring to the much more general issue of the mechanisms that reduce EPSP size, including making it undetectable. We certainly agree that is a important point to make explicitly and we have added this point to our Discussion section.

11) Previous in vitro and in vivo studies have focused on layer IV itself, while this study more broadly samples deeper cortical layers. This may influence some of the results. In particular, it has been reported that thalamocortical inputs onto FS cells in layer IV tend to be stronger and more reliable than the inputs onto RS cells. The relative similarity in EPSPs reported here may result in part from undersampling layer IV.

Most of our recordings are from simple cells in L4 and a few simple cells in L6. We now make this point clearly in the Results section.

[Editors' note: further revisions were requested prior to acceptance, as described below.]

Essential revisions:Overall, the authors have addressed most previous concerns. The text is also generally more readable and clear. The manuscript now focuses on the dynamics of the LGN-V1 synapse in vivo and simply advocates that this synaptic variability enhances post-synaptic fluctuations and therefore firing rates. While a more extensive model that addressed the intriguing issue of synchrony could provide substantial insight, we agree with the authors that a full examination deserves a separate study.- There does not seem to be a clear set of criteria for unit inclusion based on cluster metrics. The plot in Figure 1—figure supplement 1 does not make it easy to evaluate the isolation distances, but some points appear to be very low.

We report the Lratio and Isolation distance in Figure 1—figure supplement 1 as requested by the reviewers, but we did not use any defined threshold on either value to include or reject clusters. In part, because there is not any agreed upon threshold values in the literature and in part because in certain cases, these metrics did not support our visual assessment of cluster quality. We therefore relied on manual cluster-cutting and visual verification and refinement.

Although we used software that had automated cluster cutting capabilities, we manually cut clusters by visualizing the cluster space across many dimensions (features of action potentials), and visually verified each cluster included in our data. In doing so, we used several criteria for cluster definition:

1) The shape and distance of the cluster relative to the ‘noise’ cluster: We only chose clusters that were visually separate from the noise cluster.

2) The distribution of inter-spike intervals: We excluded any traces with an ISI < 1 ms.

3) The shape of action potentials included in the cluster: We manually cut action potentials that did not fit the average shape.

4) The receptive field of the cluster: We excluded any clusters that did not produce a clear center-surround receptive field characteristic of LGN cells.

Having viewed and assessed each single cluster as detailed above, we are confident in the quality of our clusters and not fully confident that Lratio and Isolation distance metrics are not sometimes confounded by the shape of noise cluster or the number and relative magnitude of cells recorded on the same tetrode.

We have added the clarifications above to the text in Figure 1—figure supplement 1.

- The final paragraph of the Introduction is somewhat confusing as written. Perhaps the authors meant "high variability of the thalamocortical synapse, rather than being irrelevant system noise, does play a functional role…"?

Corrected, thanks for the clarifying suggestion.

- There are a few places where text changes have not been fully incorporated and should be fixed, e.g. subsection “Thalamocortical synapses are unreliable and generate highly variable EPSPs”, Discussion section, figure legend for Figure 1—figure supplement 1.

Corrected, thanks.

- The authors are perhaps overly modest in citing their own previous work. The Contreras group has published several other studies (2007, 2010) on the synaptic and intrinsic properties of RS and FS cells in cat visual cortex that are relevant to the current findings.

Thanks for pointing this out. We slightly revised the text to clarify our previous findings and cited the two additional papers. The text now reads:

“This suggests that undetectable single-spike EPSPSs mask differences in the size, and rate of change, of Vm fluctuations between these cortical cell types. This is consistent with previous work from our laboratory in vivo, showing different input resistance and membrane time-constant for RS and FS cells (Contreras and Palmer, 2003, Cardin et al., 2007, Cardin et al., 2010), and consistent with greater differences observed in vitro across these cell types (Hull et al., 2009, Cruikshank et al., 2007, Schiff and Reyes, 2012, Kloc et al., 2014).”

- Subsection “Thalamocortical synapses are unreliable and generate highly variable EPSPs”, "…there is a population of very large (>4 mV) EPSPs triggered by LGN input." This unusual population is interesting and warrant comparison with the more typical 0.5-1.0 mV EPSPs. Most of the traces shown are the common small type. Figure 3B green appears to show a larger one that is 2-3 mV at most. These very large EPSPs could result from nonlinear voltage-dependent mechanisms that might have a different shape from normal EPSPs, bursts of LGN spikes, etc. The green traces indeed exhibit a long shoulder suggestive of voltage-dependent mechanisms. Peak normalizing the average small and large EPSPs should allow visual comparison of their shapes.

We had analyzed the shape, and specifically the slope, of these rare large EPSPs and had not found significant differences. Thanks for the reminder to include this negative result, we have updated Figure 3—figure supplement 3 to include peak-normalized traces of the largest, and more frequent small, single-spike EPSPs. The shape of the small and large EPSPs is similar.

- On a related note, subsection “Detecting Single-spike EPSPs” states that small "undetectable" EPSPs are unlikely to influence postsynaptic Vm. While I agree with this statement generally, I think the authors have to qualify it with "at least under our conditions". One can easily imagine nonlinear voltage-gated channels that make such small EPSPs potent under different conditions (e.g., state dependent neuromodulation of those channels, disinhibitory circuits, etc.). Also, the interim results summary paragraph, subsection “subsection “Thalamocortical synapses are unreliable and generate highly variable EPSPs”, is confusing. Consider "…including a large percentage of EPSPs that do not visibly affect somatic Vm and are unlikely to contribute to the cortical output." As noted in point 2, this is not guaranteed. More so it doesn't seem to allow for the possibility of synaptic failures. This and the surrounding text could be read to mean they "undetected" EPSPs are always small and never failures.

We added, ‘at least under our conditions.’ We fully agree that some undetectable EPSPs, perhaps the majority, may be failures. However, since we can’t know this, we were being cautious with our terminology. We have included the possibility for failures. The text now reads:

“In summary, we found that thalamocortical synapses in the cat visual system produce EPSPs with small and variable amplitudes, including a large percentage of undetectable EPSPs, thalamic spikes that either fail to produce an EPSP or produce an EPSP too small to visibly affect somatic Vm.”

The authors have improved the short-term depression section, but there is still some difficulty following the logic of the predictions (subsection “Short-term synaptic plasticity in vivo”). Boudreau and Ferster, 2005 electrically stimulated LGN inputs in anesthetized cats and predicted that depression would be near saturated in the awake animal. The present study, also under anesthesia, evoked LGN spikes by visual stimulation and measured the synaptic input. They find that LGN firing rate does not correlate with the amount of depression (short vs long ISI) they observe, which in any case is about a 17% decrease. However, Boudreau and Ferster's depression curve does show a slight similar decrease from the first to later stimuli. Additionally, the present study only has 3 data points above 10 Hz-how can it be used to make claims about what happens for awake firing rates? 1-10 Hz is already in the range of awake spontaneous activity.

We apologize but we do not understand what the reviewer means here.

First, our average value of STSD (short-term synaptic depression) is -32 +- 15%. The value of 17% indicated by the reviewer is a population average that also includes some synapses with short-term synaptic facilitation (STSF), which in our nomenclature is expressed as a positive number and thus reduces the average from -32 to -17. We now realize that this population number is confusing and clarified the text by reporting averages for synapses exhibiting significant STSD and STSF separately. The text now reads:

“The% change across the 17 connected pairs exhibiting STSD was -32 ± 15%, and the% change across the 3 connected pairs exhibiting STSF was 24 ± 16%.”

Second, we show 9 datapoints above 10Hz in Figure 5H and not 3 as indicated by the reviewer, and our range of firing rates in the anesthetized cat are similar to those reported by Boudreau and Ferster.

Boudreau and Ferster report that short-term depression, calculated using response to electrical stimulation, is significantly higher at LGN firing rates of 4.1 ± 1.4 spikes/sec compared to firing rates of 11.8 ± 2.9 spikes/sec. Both of these were recorded in the anesthetized cat, with the lower rate measured under artificially increased intraocular pressure. They use the fact that short-term depression is lower at 11 Hz to hypothesize that at the even higher firing rate of visually driven activity in the awake cat (they cite 20-40 Hz), short-term depression would be saturated and potentially not detectable.

Our observations are different. We have a similar range of LGN firing rates in the anesthetized cat (Figure 5H, most cells~1-12 Hz). However, unlike Boudreau and Ferster, we do not see a relationship between the amount of short-term depression and LGN firing rate in this range. Because our data do not support a relationship between firing rate and short-term depression, we argue against the hypothesis that short-term depression would be saturated at the higher firing rates in the awake condition.